# Causal Strategic Inference in Networked Microfinance Economies

**Mohammad T. Irfan**
Department of Computer Science
Bowdoin College
Brunswick, ME 04011
mirfan@bowdoin.edu

**Luis E. Ortiz**
Department of Computer Science
Stony Brook University
Stony Brook, NY 11794
leortiz@cs.stonybrook.edu

## Abstract

Performing interventions is a major challenge in economic policy-making. We propose *causal strategic inference* as a framework for conducting interventions and apply it to large, networked microfinance economies. The basic solution platform consists of modeling a microfinance market as a networked economy, learning the parameters of the model from the real-world microfinance data, and designing algorithms for various causal questions. For a special case of our model, we show that an equilibrium point always exists and that the equilibrium interest rates are unique. For the general case, we give a constructive proof of the existence of an equilibrium point. Our empirical study is based on the microfinance data from Bangladesh and Bolivia, which we use to first learn our models. We show that causal strategic inference can assist policy-makers by evaluating the outcomes of various types of interventions, such as removing a loss-making bank from the market, imposing an interest rate cap, and subsidizing banks.

## 1 Introduction

Although the history of microfinance systems takes us back to as early as the 18th century, the foundation of the modern microfinance movement was laid in the 1970s by Muhammad Yunus, a then-young Economics professor in Bangladesh. It was a time when the newborn nation was struggling to recover from a devastating war and an ensuing famine. A blessing in disguise may it be called, it led Yunus to design a small-scale experimentation on micro-lending as a tool for poverty alleviation. The feedback from that experimentation gave Yunus and his students the insight that micro-lending mechanism, with its social and humanitarian goals, could successfully intervene in the informal credit market that was predominated by opportunistic moneylenders. Although far from experiencing a smooth ride, the microfinance movement has nevertheless been a great success story ever since, especially considering the fact that it began with just a small, out-of-pocket investment on 42 clients and boasts a staggering 100 million poor clients worldwide at present [27]. Yunus and his organization Grameen Bank have recently been honored with the Nobel peace prize "for their efforts to create economic and social development from below."

A puzzling element in the success of microfinance programs is that while commercial banks dealing with well-off customers struggle to recover loans, microfinance institutions (MFI) operate without taking any collateral and yet experience very low default rates! The central mechanism that MFIs use to mitigate risks is known as the *group lending with joint-liability contract*. Roughly speaking, loans are given to groups of clients, and if a person fails to repay her loan, then either her partners repay it on her behalf or the whole group gets excluded from the program. Besides risk-mitigation, this mechanism also helps lower MFI's cost of monitoring clients' projects. Group lending with a joint-liability contract also improves repayment rates and mitigates *moral hazard* [13]. Group lending and many other interesting aspects of microfinance systems, such as efficiency and distribution of

intervening informal credit markets, failure of pro-poor commercial banks, gender issues, subsidies, etc., have been beautifully delineated by de Aghion and Morduch in their book [9].

Here, we assume that assortative matching and joint-liability contracts would mitigate the risks of adverse selection [13] and moral hazard. We further assume that due to these mechanisms, there would be no default on loans. This assumption of complete repayment of loans may seem to be very much idealistic. However, practical evidence suggests very high repayment rates. For example, Grameen Bank's loan recovery rate is 99.46% [21].

Next, we present causal strategic inference, followed by our model of microfinance markets and our algorithms for computing equilibria and learning model parameters. We present an empirical study at the end. We leave much of the details to the Appendix, located in the supplementary material.

## 2 Causality in Strategic Settings

Going back two decades, one of the most celebrated success stories in the study of causality, which studies cause and effect questions using mathematical models of real-world phenomena, was the development of causal probabilistic inference. It was led by Judea Pearl, who was later awarded the ACM Turing prize in 2011 for his seminal contribution. In his highly acclaimed book on causality, Pearl organizes causal queries in probabilistic settings in three different levels of difficulty— prediction, interventions, and counterfactuals (in the order of increasing difficulty) [22, p. 38]. For example, an intervention query is about the effects of changing an existing system by what Judea Pearl calls "surgery." We focus on this type of query here.

**Causal Strategic Inference.** We study causal inferences in *game-theoretic settings* for intervention-type queries. Since game theory reliably encodes strategic interactions among a set of players, we call this type of inference *causal strategic inference*. Note that interventions in game-theoretic settings are not new (see Appendix B for a survey). Therefore, we use causal strategic inference simply as a convenient name here. Our main contribution is a framework for performing causal strategic inference in networked microfinance economy.

As mentioned above, interventions are carried out by surgeries. So, what could be a surgery in a game-theoretic setting? Analogous to the probabilistic settings [22, p. 23], the types of surgeries we consider here change the "structure" of the game. This can potentially mean changing the payoff function of a player, removing a player from the game, adding a new player to the game, changing the set of actions of a player, as well as any combination of these. We discuss other possibilities in Appendix A. See also [14].

The proposed framework of causal strategic inference is composed of the following components: mathematically *modeling* a complex system, *learning* the parameters of the model from *real-world data*, and designing *algorithms* to predict the effects of interventions.

**Review of Literature.** There is a growing literature in econometrics on modeling strategic scenarios and estimating the parameters of the model. Examples are Bjorn and Vuong's model of labor force participation [5], Bresnahan and Reiss' entry models [6, 7], Berry's model of airline markets [4], Seim's model of product differentiation [24], Augereau et al.'s model of technology adoption [2]. A survey of the recent results is given by Bajari et al. [3]. All of the above models are based on McFadden's random utility model [18], which often leads to an analytical solution. In contrast, our model is based on classical models of two-sided economies, for which there is no known analytical solution. Therefore, our solution approach is algorithmic, not analytic.

More importantly, although all of the above studies model a strategic scenario and estimate the parameters of the respective model, *none of them perform any intervention, which is one of our main goals*. We present more details on each of these as well as several additional studies in Appendix B.

Our model is closely related to the classical Fisher model [12]. An important distinction between our model and Fisher's, including its graphical extension [16], is that our model allows buyers (i.e., villages) to invest the goods (i.e., loans) in productive projects, thereby generating revenue that can be used to pay for the goods (i.e., repay the loans). In other words, the crucial modeling parameter of "endowment" is no longer a constant in our case. For the same reason, the classical Arrow-Debreu model [1] or the recently developed graphical extension to the Arrow-Debreu model [15], does not capture our setting. Moreover, in our model, the buyers have a very different objective function.

# 3 Our Model of Microfinance Markets

We model a microfinance market as a two-sided market consisting of MFIs and villages. Each MFI has branches in a subset of the villages, and each branch of an MFI deals with the borrowers in that village only. Similarly, each village can only interact with the MFIs present there.

We use the following notation. There are $n$ MFIs and $m$ villages. $V_i$ is the set of villages where MFI $i$ operates and $B_j$ is the set of MFIs that operate in village $j$. $T_i$ is the finite total amount of loan available to MFI $i$ to be disbursed. $g_j(l) := d_j + e_j l$ is the revenue generation function of village $j$ (parameterized by the loan amount $l$), where the initial endowment $d_j > 0$ (i.e., each village has other sources of income [9, Ch. 1.3]) and the rate of revenue generation $e_j \geq 1$ are constants. $r_i$ is the flat interest rate of MFI $i$ and $x_{j,i}$ is the amount of loan borrowed by village $j$ from MFI $i$. Finally, the villages have a *diversification parameter* $\lambda \geq 0$ that quantifies how much they want their loan portfolios to be diversified. [1] The problem statement is given below.

Following are the inputs to the problem. First, for each MFI $i$, $1 \leq i \leq n$, we are given the total amount of money $T_i$ that the MFI has and the set $V_i$ of villages that the MFI has branches. Second, for each village $j$, $1 \leq j \leq m$, we are given the parameters $d_j > 0$ and $e_j > 1$ of the village's revenue generation function [2] and the set $B_j$ of the MFIs that operate in that village.

*MFI-side optimization problem.* Each MFI $i$ wants to set its interest rate $r_i$ such that all of its loan is disbursed. This is known as *market-clearance* in economics. Here, the objective function is a constant due to the MFIs' goal of market-clearance.

$$\max_{r_i} \quad 1$$

$$\text{subject to} \quad r_i \left( T_i - \sum_{j \in V_i} x_{j,i} \right) = 0$$

$$\sum_{j \in V_i} x_{j,i} \leq T_i \qquad (P_M)$$

$$r_i \geq 0$$

*Village-side optimization problem.* Each village $j$ wants to maximize its diversified loan portfolio, subject to its repaying it. We call the second term of the objective function of $(P_V)$ the *diversification term*, where $\lambda$ is chosen using the data. [3] We call the first constraint of $(P_V)$ the *budget constraint*.

$$\max_{\mathbf{x}_j = (x_{j,i})_{i \in B_j}} \quad \sum_{i \in B_j} x_{j,i} + \lambda \sum_{i \in B_j} x_{j,i} \log \frac{1}{x_{j,i}}$$

$$\text{subject to} \quad \sum_{i \in B_j} x_{j,i}(1 + r_i - e_j) \leq d_j \qquad (P_V)$$

$$\mathbf{x}_j \geq 0$$

For this two-sided market, we use an *equilibrium point* as the solution concept. It is defined by an interest rate $r_i^*$ for each MFI $i$ and a vector $\mathbf{x}_j^* = (x_{j,i}^*)_{i \in B_j}$ of loan allocations for each village $j$ such that the following two conditions hold. First, given the allocations $\mathbf{x}^*$, each MFI $i$ is optimizing the program $(P_M)$. Second, given the interest rates $\mathbf{r}^*$, each village $j$ is optimizing the program $(P_V)$.

**Justification of Modeling Aspects.** Our model is inspired by the book of de Aghion and Morduch [9] and several other studies [20, 26, 23]. We list some of our modeling aspects below.

*Objective of MFIs.* It may seem unusual that although MFIs are banks, we do not model them as profit-maximizing agents. The perception that MFIs make profits while serving the poor has been described as a "myth" [9, Ch. 1]. In fact, the book devotes a whole chapter to bust this myth [9, Ch. 9]. Therefore, empirical evidence supports modeling MFIs as not-for-profit organizations.

*Objective of Villages.* Typical customers of MFIs are low-income people engaged in small projects and most of them are women working at home (e.g., Grameen Bank has a 95% female customer base) [9]. Clearly, there is a distinction between customers borrowing from an MFI and those borrowing from commercial banks. Therefore, we model the village side as non-corporate agents.

*Diversification of Loan Portfolios.* Empirical studies suggest that the village side does not maximize its loan by borrowing only from the lowest interest rate MFI [26, 23]. There are other factors, such as large loan sizes, shorter waiting periods, and flexible repayment schemes [26]. We added the diversification term in the village objective function to reflect this. Furthermore, this formulation is in line with the quantal response approach [19] and human subjects are known to respond to it[17].

*Complete repayment of loans.* A hallmark of microfinance systems worldwide is very high repayment rates. For example, the loan recovery rate of Grameen Bank is 99.46% and PKSF 99.51% [21]. Due to such empirical evidence, we assume that the village-side completely repays its loan.

## 3.1 Special Case: No Diversification of Loan Portfolios

It will be useful to first study the case of non-diversified loan portfolios, i.e., $\lambda = 0$. In this case, the villages simply wish to maximize the amount of loan that they can borrow. Several properties of an equilibrium point can be derived for this special case. We give the complete proofs in Appendix C.

**Property 3.1.** *At any equilibrium point* $(\mathbf{x}^*, \mathbf{r}^*)$, *every MFI* $i$'s *supply must match the demand for its loan, i.e.,* $\sum_{j \in V_i} x_{j,i}^* = T_i$. *Furthermore, every village* $j$ *borrows only from those MFIs* $i \in B_j$ *that offer the lowest interest rate. That is,* $\sum_{i \in B_j, r_i^* = r_{m_j}^*} x_{j,i}^*(1 + r_i^* - e_j) = d_j$ *for any MFI* $m_j \in \arg\min_{i \in B_j} r_i^*$, *and* $x_{j,k}^* = 0$ *for any MFI* $k$ *such that* $r_k^* > r_{m_j}^*$.

*Proof Sketch.* Show by contradiction that at an equilibrium point, the constraints of the village-side or the MFI-side optimization are violated otherwise. □

We next present a lower bound on interest rates at an equilibrium point.

**Property 3.2.** *At any equilibrium point* $(\mathbf{x}^*, \mathbf{r}^*)$, *for every MFI* $i$, $r_i^* > \max_{j \in V_i} e_j - 1$.

*Proof Sketch.* Otherwise, the village-side demand would be unbounded, which would violate the MFI-side constraint $\sum_{j \in V_i} x_{j,i}^* \leq T_i$. □

Following are two related results that preclude certain trivial allocations such as all the allocations being zero at an equilibrium point.

**Property 3.3.** *At any equilibrium point* $(\mathbf{x}^*, \mathbf{r}^*)$, *for any village* $j$, *there exists an MFI* $i \in B_j$ *such that* $x_{j,i}^* > 0$.

*Proof Sketch.* In this case, $j$ satisfies its constraints but does not maximize its objective function. □

**Property 3.4.** *At any equilibrium point* $(\mathbf{x}^*, \mathbf{r}^*)$, *for any MFI* $i$, *there exists a village* $j \in V_i$ *such that* $x_{j,i}^* > 0$.

*Proof Sketch.* The first constraint of $(P_M)$ for MFI $i$ is violated. □

## 3.2 Eisenberg-Gale Formulation

We now present an Eisenberg-Gale convex program formulation of a restricted case of our model where the diversification parameter $\lambda = 0$ and all the villages $j$, $1 \leq j \leq m$, have the same revenue generation function $g_j(l) := d + el$, where $d > 0$ and $e \geq 1$ are constants. We first prove that this case is equivalent to the following Eisenberg-Gale convex program [11, 25], which gives us the existence of an equilibrium point and the uniqueness of the equilibrium interest rates as a corollary. Below is the Eisenberg-Gale convex program [11, p. 166].

$$\min_{\mathbf{z}} \quad \sum_{j=1}^{m} -\log \sum_{i \in B_j} z_{j,i}$$

$$\text{subject to} \quad \sum_{j \in V_i} z_{j,i} - T_i \le 0, \quad 1 \le i \le n \qquad (P_E)$$

$$z_{j,i} \ge 0, \qquad\qquad 1 \le i \le n, \; j \in V_i$$

We have the following theorem and corollary.

**Theorem 3.5.** *The special case of microfinance markets with identical villages and no loan portfolio diversification, has an equivalent Eisenberg-Gale formulation.*

*Proof Sketch.* The complete proof is very long and given in Appendix C. We first make a connection between an equilibrium point $(\mathbf{x}^*, \mathbf{r}^*)$ of a microfinance market and the variables of program $(P_E)$. In particular, we define $x_{j,i}^* \equiv z_{j,i}^*$ and express $r_i^*$ in terms of certain dual variables of $(P_E)$. Using the properties given in Section 3.1, we show that the equilibrium conditions of $(P_M)$ and $(P_V)$ in this special case are equivalent to the Karush-Kuhn-Tucker (KKT) conditions of $(P_E)$. $\qquad \square$

**Corollary 3.6.** *For the above special case, there exists an equilibrium point with unique interest rates [11] and a combinatorial polynomial-time algorithm to compute it [25].*

An implication of Theorem 3.5 is that in a more restricted case of our model (with the additional constraint of $T_i$ being same for all MFI $i$), our model is indeed a graphical linear Fisher model where all the "utility coefficients" are set to $1$ (see the convex program 5.1 [25] to verify this).

### 3.3 Equilibrium Properties of General Case

In the general case, the objective function of $(P_V)$ can be written as $\sum_{i \in B_j} x_{j,i} - \lambda \sum_{i \in B_j} x_{j,i} \log x_{j,i}$. While the first term wants to maximize the total amount of loan, the second (diversification term) wants, in colloquial terms, "not to put all the eggs in one basket." If $\lambda$ is sufficiently small, then the first term dominates the second, which is a desirable assumption.

**Assumption 3.1.**

$$0 \le \lambda \le \frac{1}{2 + \log T_{max}}$$

*where $T_{max} \equiv \max_i T_i$ and w.l.o.g., $T_i > 1$ for all $i$.*

The following equilibrium properties will be used in the next section.

**Property 3.7.** *The first constraint of $(P_V)$ must be tight at any equilibrium point.*

*Proof Sketch.* Otherwise, the village can increase its objective function slightly. $\qquad \square$

We define $e_{max}^i \equiv \max_{j \in V_i} e_j$ and $d_{max}^i \equiv \max_{j \in V_i} d_j$ and obtain the following bounds.

**Property 3.8.** *At any equilibrium point, for each MFI $i$, $e_{max}^i - 1 < r_i^* \le \frac{|V_i| d_{max}^i}{T_i} + e_{max}^i - 1$.*

*Proof Sketch.* The proof of $e_{max}^i - 1 < r_i^*$ is similar to the proof of Property 3.2. The upper bound is derived from the maximum loan a village $j$ can seek from the MFI $i$ at an equilibrium point. $\qquad \square$

## 4 Computational Scheme

For the clarity of presentation we first design an algorithm for equilibrium computation and then talk about learning the parameters of our model.

### 4.1 Computing an Equilibrium Point

We give a constructive proof of the existence of an equilibrium point in the microfinance market defined by $(P_M)$ and $(P_V)$. The inputs are $\lambda > 0$, $e_j$ and $d_j$ for each village $j$, and $T_i$ for each MFI $i$. We first give a brief outline of our scheme in Algorithm 1.

---

**Algorithm 1** Outline of Equilibrium Computation

---

1: For each MFI $i$, initialize $r_i$ to $e_{max}^i - 1$.
2: For each village $j$, compute its best response $\mathbf{x}_j$.
3: **repeat**
4:     **for all** MFI $i$ **do**
5:         **while** $T_i \neq \sum_{j \in V_i} x_{j,i}$ **do**
6:             Change $r_i$ as described after Lemma 4.3.
7:             For each village $j \in V_i$, update its best response $\mathbf{x}_j$ reflecting the change in $r_i$.
8:         **end while**
9:     **end for**
10: **until** no change to $r_i$ occurs for any $i$

---

Before going on to the details of how to change $r_i$ in Line 6 of Algorithm 1, we characterize the best response of the villages used in Line 7.

**Lemma 4.1.** *(Village's Best Response) Given the interest rates of all the MFIs, the following is the unique best response of any village $j$ to any MFI $i \in B_j$:*

$$x_{j,i}^* = \exp\left(\frac{1 - \lambda - \alpha_j^*(1 + r_i - e_j)}{\lambda}\right) \tag{1}$$

*where $\alpha_j^* \geq 0$ is the unique solution to*

$$\sum_{i \in B_j} \exp\left(\frac{1 - \lambda - \alpha_j^*(1 + r_i - e_j)}{\lambda}\right)(1 + r_i - e_j) = d_j. \tag{2}$$

*Proof Sketch.* Derive the Lagrangian of $(P_V)$ and argue about optimality. $\square$

Therefore, as soon as $r_i$ of some MFI $i$ changes in Line 6 of Algorithm 1, both $x_{j,i}^*$ and the Lagrange multiplier $\alpha_j^*$ change in Line 7, for any village $j \in V_i$. Next, we show the direction of these changes.

**Lemma 4.2.** *Whenever $r_i$ increases (decreases) in Line 6, $x_{j,i}$ must decrease (increase) for every village $j \in V_i$ in Line 7 of Algorithm 1.*

*Proof Sketch.* Rewrite the expression of $x_{j,i}^*$ given in Lemma 4.1 in terms of $\alpha_j^*$. Do the same for $x_{j,k}^*$ for some $k \in B_j$. Use the two expressions for $\alpha_j^*$ to argue about the increase of $r_i$. $\square$

The next lemma is a cornerstone of our theoretical results. Here, we use the term *turn of an MFI* to refer to the iterative execution of Line 6, wherein an MFI sets its interest rate to clear its market.

**Lemma 4.3.** *(Strategic Complementarity) Suppose that an MFI $i$ has increased its interest rate at the end of its turn. Thereafter, it cannot be the best response of any other MFI $k$ to lower its interest rate when its turn comes in the algorithm.*

*Proof Sketch.* The proof follows from Lemma 4.2 and Assumption 3.1. The main task is to show that when $r_i$ increases $\alpha_j^*$ for $j \in V_i$ cannot increase. $\square$

In essence, Lemma 4.2 is a result of *strategic substitutability* [10] between the MFI and the village sides, while Lemma 4.3 is a result of *strategic complementarity* [8] among the MFIs. Our algorithm exploits these two properties as we fill in the details of Lines 6 and 7 next.

*Line 6: MFI's Best Response.* By Lemma 4.2, the total demand for MFI $i$'s loan monotonically decreases with the increase of $r_i$. We use a binary search between the upper and the lower bounds of $r_i$ given in Property 3.8 to find the "right" value of $r_i$. More details are given in Appendix D.

*Line 7: Village's Best Response.* We use Lemma 4.1 to compute each village $j$'s best response $x_{j,i}^*$ to MFIs $i \in B_j$. However, Equation (1) requires computation of $\alpha_j^*$, the solution to Equation (2). We exploit the convexity of Equation (2) to design a simple search algorithm to find $\alpha_j^*$.

**Theorem 4.4.** *There always exists an equilibrium point in a microfinance market specified by programs $(P_M)$ and $(P_V)$.*

*Proof Sketch.* Use Lemmas 4.3 and 4.1 and the well-known monotone convergence theorem. $\square$

### 4.2 Learning the Parameters of the Model

The inputs are the spatial structure of the market, the observed loan allocations $\tilde{x}_{j,i}$ for all village $j$ and all MFI $i \in B_j$, the observed interest rates $\tilde{r}_i$ and total supply $T_i$ for all MFI $i$. The objective of the learning scheme is to instantiate parameters $e_j$ and $d_j$ for all $j$. We learn these parameters using the program below so that an equilibrium point closely approximates the observed data.

$$
\min_{\mathbf{e},\mathbf{d},\mathbf{r}} \sum_i \sum_{j \in V_i} (x^*_{j,i} - \tilde{x}_{j,i})^2 + C \sum_i (r^*_i - \tilde{r}_i)^2
$$

such that

for all $j$,

$$
\mathbf{x}^*_j \in \arg\max_{\mathbf{x}_j} \sum_{i \in B_j} x_{j,i} + \lambda \sum_{i \in B_j} x_{j,i} \log \frac{1}{x_{j,i}}
$$

$$
\text{s. t.} \sum_{i \in B_j} x_{j,i}(1 + r^*_i - e_j) \leq d_j
$$

$$
\mathbf{x}_j \geq 0 \tag{3}
$$

$$
\mathbf{e}_j \geq 1, \ \mathbf{d}_j \geq 0
$$

$$
\sum_{j \in V_i} x^*_{j,i} = T_i, \ \text{for all } i
$$

$$
r_i \geq e_j - 1, \ \text{for all } i \text{ and all } j \in V_i
$$

The above is a nested (bi-level) optimization program. The term $C$ is a constant. In the interior optimization program, $\mathbf{x}^*$ are best responses of the villages, w.r.t. the parameters and the interest rates $\mathbf{r}^*$. In practice, we exploit Lemma 4.1 to compute $\mathbf{x}^*$ more efficiently, since it suffices to search for Lagrange multipliers $\alpha_j$ in a much smaller search space and then apply Equation (1). We use the interior-point algorithm of Matlab's large-scale optimization package to solve the above program. In the next section, we show that the above learning procedure does not overfit the real-world data. We also highlight the issue of equilibrium selection for parameter estimation.

## 5 Empirical Study

We now present our empirical study based on the microfinance data from Bolivia and Bangladesh. The details of this study can be found in Appendix E (included in the supplementary material).

**Case Study: Bolivia**

*Data.* We obtained microfinance data of Bolivia from several sources, such as ASOFIN, the apex body of MFIs in Bolivia, and the Central Bank of Bolivia. [4] We were only able to collect somewhat coarse, region-level data (June 2011). It consists of eight MFIs operating in 10 regions.

*Computational Results.* We first choose a value of $\lambda$ such that the objective function value of the learning optimization is low as well as "stable" and the interest rates are also relatively dissimilar. Using this value of $\lambda$, the learned $e_j$'s and $d_j$'s capture the variation among the villages w.r.t. the revenue generation function. The learned loan allocations closely approximate the observed allocations. The learned model matches each MFI's total loan allocations due to the learning scheme.

*Issues of Bias and Variance.* Our dataset consists of a single sample. As a result, the traditional approach of performing cross validation using hold-out sets or plotting learning curves by varying the number of samples do not work in our setting. Instead, we systematically introduce noise to the observed data sample. In the case of overfitting, increasing the level of noise would lead the equilibrium outcome to be significantly different from the observed data. To that end, we used two noise models–Gaussian and Dirichlet. In both cases, the training and test errors are very low and the learning curves do not suggest overfitting.

*Equilibrium Selection.* In the case of multiple equilibria, our learning scheme biases its search for an equilibrium point that most closely explains the data. However, does the equilibrium point change drastically when noise is added to data? For this, we extended the above procedure using a bootstrapping scheme to measure the distance between different equilibrium points when noise is added. For both Gaussian and Dirichlet noise models, we found that the equilibrium point does not change much even with a high degree of noise. Details, including plots, are given in Appendix E.

**Case Study: Bangladesh**

Based on the microfinance data (consisting of seven MFIs and 464 villages/regions), dated December 2005, from Palli Karma Sahayak Foundation (PKSF), which is the apex body of NGO MFIs in Bangladesh, we have obtained very similar results to the Bolivia case (see Appendix E).

# 6   Policy Experiments

For a specific intervention policy, e.g., removal of government-owned MFIs, we first learn the parameters of the model and then compute an equilibrium point, both in the original setting (before removal of any MFI). Using the parameters learned, we compute a new equilibrium point after the removal of the government-owned MFIs. Finally, we study changes in these two equilibria (before and after removal) in order to predict the effect of such an intervention.

*Role of subsidies.* MFIs are very much dependent on subsidies [9, 20]. We ask a related question: how does giving subsidies to an MFI affect the market? For instance, one of the Bolivian MFIs named Eco Futuro exhibits very high interest rates both in observed data and at an equilibrium point. Eco Futuro is connected to all the villages, but has very little total loan to be disbursed compared to the leading MFI Bancosol. Using our model, if we inject further subsidies into Eco Futuro to make its total loan amount equal to Bancosol's, not only do these two MFIs have the same (but lower than before) equilibrium interest rates, it also drives down the interest rates of the other MFIs.

*Changes in interest rates.* Our model computes lower equilibrium interest rate (around 12%) for ASA than its observed interest rate (15%). It is interesting to note that in late 2005, ASA lowered its interest rate from 15% to 12.5%, which is close to what our model predicts at an equilibrium point. [5]

*Interest rates ceiling.* PKSF recently capped the interest rates of its partner organization to 12.5% [23], and more recently, the country's Microfinance Regulatory Authority has also imposed a ceiling on interest rate at around 13.5% flat. [6] Such evidence on interest rate ceiling is consistent with the outcome of our model, since in our model, $13.4975\%$ is the highest equilibrium interest rate.

*Government-owned MFIs.* Many of the government-owned MFIs are loss-making [26]. Our model shows that removing government-owned MFIs from the market would result in an increase of equilibrium interest rates by approximately $0.5\%$ for every other MFI. It suggests that less competition leads to higher interest rates, which is consistent with empirical findings [23].

*Adding new branches.* Suppose that MFI Fassil in Bolivia expands its business to all villages. It may at first seem that due to the increase in competition, equilibrium interest rates would go down. However, since Fassil's total amount of loan does not change, the new connections and the ensuing increase in demand actually increase equilibrium interest rates of all MFIs.

*Other types of intervention.* Through our model, we can ask more interesting questions such as would an interest rate ceiling be still respected after the removal of certain MFIs from the market? Surprisingly, according to our discussion above, the answer is yes if we were to remove government-owned MFIs. Similarly, we can ask what would happen if a major MFI gets entirely shut down? We can also evaluate effects of subsidies from the donor's perspective (e.g., which MFIs should a donor select and how should the donor distribute its grants among these MFIs in order to achieve some goal). Causal questions like these form the long-term goal of this research.

**Acknowledgement**

We thank the reviewers. Luis E. Ortiz was supported in part by NSF CAREER Award IIS-1054541.

## Footnotes

[1] For simplicity, we assume that all the villages have the same diversification parameters.

[2] When we apply our model to real-world settings, we will see that in contrast to the other inputs, $d_j$ and $e_j$ are not explicitly mentioned in the data and therefore, need to be learned from the data. The machine learning scheme for that will be presented in Section 4.2.

[3] Note that although this term bears a similarity with the well-known entropic term, they are different, because $x_{j,i}$'s here can be larger than 1.

[4]http://www.asofinbolivia.com; http://www.bcb.gob.bo/

[5] http://www.adb.org/documents/policies/ microfinance/microfinance0303.asp?p=microfnc.

[6] http://www.microfinancegateway.org/ p/site/m/template.rc/1.1.10946/

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
