[Supplementary Material]

# Appendix:
# Causal Strategic Inference in Networked Microfinance Economies

## A    Causality

Causality is one of the most natural quests of the human mind. Not only that it appears in abundance in our daily life, it also has a long history of scientific expedition, often embroiled in debates among statisticians, philosophers, economists, and computer scientists [38, 47, 30]. Such a level of contention among researchers of diverse backgrounds *on one single topic* is rare and at the same time indicative of its scientific import and wide applicability. This paper presents a comprehensive framework for studying causality in *strategic* scenarios that often appear in economic settings.

One beautiful example of causality being a contested ground is a book edited by Daniel Little [38]. The chapters of the book pave the way for an enlightening back and forth debate between philosophers and economists. For example, in Chapter 2, philosopher James Woodward presents a causal interpretation of the structural equation models frequently used by economists. Woodward promotes the *manipulability theory* of causation as opposed to other alternatives, such as Granger's notion of causation [28]. The manipulability theory resonates with our intuitive perception of causation. That is, if one variable causes another in a relationship, then changing or *intervening* the first variable (or other related variables) would provide a way of manipulating the latter. Now, an important question is: does the relationship remain stable while these interventions are being made? In the case of an *autonomous* relationship, the answer is yes. One example of an autonomous relationship is a law of the physics, such as the law of gravitation, which remains valid under a wide range of interventions. In contrast, non-autonomous relationships would break down easily under slight changes.

However, instead of thinking of relationships as simply autonomous versus non-autonomous, Woodward suggests the notion of the *degree of au-*

*tonomy*, which corresponds to the range of interventions (perhaps limited) under which a relationship would remain stable. Woodward argues that this notion is particularly well suited for interpreting structural equation models. One significance of this is that it gives these models an explanatory power, as Woodward says, "autonomous relationships are causal in character and can be used to provide explanations."

Later on, in Chapter 4 of the book, economist Kevin Hoover presents his view of causality in econometrics while contesting various points made by the authors of the earlier two chapters, including Woodward [38]. To a large degree, Hoover's view concurs with that of Woodward. However, the two disagree on some of the fundamental issues, such as the explanatory power of a causal relationship. Hoover contends that "econometric models do not explain." Hoover also contests many of the finer constructs, such as the meanings of "law" and "theory" implied by Woodward, in contrast to an econometrician's interpretation of these terms.

The reason we brought up the debate between philosophers and economists is two-fold. First, it gives a snapshot of the ever-contested topic of causality, which only highlights its importance across various disciplines. Second, it exposes a key component of the causality study in general—interventions or changes made to a system. We will illustrate how we incorporate interventions in strategic settings, but first, we will take a detour to some recent happenings in order to further signify the notion of interventions in causality studies.

## A.1   Causal Probabilistic Inference

In recent times, one of the most celebrated success stories in the study of causality is the development of causal probabilistic inference during the 1990s [48, 49, 45, 50, 46, 47]. Applications of causal probabilistic inference can now be seen in very diverse disciplines, such as economics, public policy, sociology, computer science, and various branches of life sciences, to name just a few. Given its emergence in wide-ranging application domains, it may at first be surprising to learn that the issue of causation has been swept under the rug for decades in classical statistics until 1935 when Sir Ronald Fisher's seminal work on randomized experiments [24, 25] was published [47, p. 339–342]. Correlation, rather than causation, had been the prescriptive concept in statistics all those years. However, correlation alone does not directly answer questions such as: Does smoking *cause* cancer? Or, will increasing taxes *cause* the national debt to go down?

Judea Pearl, the recipient of the ACM Turing award in 2012 and one of

the forerunners in the pursuit of studying causality in probabilistic settings, notes that the reason for this apparent neglect of causation in classical statistics is deeply rooted in the inability of probability theory to express causal statements [47, p. 342]. In particular, the language of probability theory is geared toward expressing *observational* inferences, as opposed to causal ones. In an observational inference, we may seek the probability of some events happening *given that* some other events have happened. Probability theory lays out a clear set of rules on how to express and manipulate such an inference question in order to give an answer to this question. In contrast to observational inferences where some events are observed (or given), causal inferences are accompanied by the mechanism of *intervention*. An example of an intervention is to set a random variable $X$ (over which we have control) to a specific value $x$, which Pearl denotes by $do(X = x)$ [47, p. 23]. An inference question in connection with this intervention would be to ask what would the probability of some events happening be once we perform this *do* operation.

On the surface, the causal inference question having a $do(X = x)$ operation may seem to be very similar to an observational inference question where $X = x$ is given. However, there are two notable differences. First, the *do* operation has the power to change the dependency structure among the random variables. Therefore, it can potentially change the joint probability distribution of the random variables. Second and as alluded above, probability theory, in its originality, cannot express this *do* operation mathematically. The study of causality in probabilistic settings has provided us with a mathematical framework that extends probability theory to express and process interventions.

In fact, Judea Pearl takes a broader view of causality than just the *do* operation. According to him, the study of causality can be hierarchically organized in three natural types of queries with increasing levels of difficulty: predictions, interventions, and counterfactuals [47, p. 38]. First, prediction is the type of query where we observe something about the "system," and taking that observational knowledge into account, we are asked to infer something else that we did not observe. The important aspect in prediction is that we are not allowed to change anything in the system. Changing something in the system, which Judea Pearl often refers to as *surgery*, is permitted in the second type of causal query—interventions. An example is the the *do* operation in probabilistic settings, as outlined above. Counterfactuals, the third type of causal queries, are the most challenging ones in the sense that we are given some observation about the system and asked to infer the outcome of the system if the opposite of that observation, in some

sense, were to take place.

The goal of this paper is to study causal inferences in game-theoretic settings at the second level of queries, interventions. Since game theory reliably encodes strategic interactions among a set of players, we will call this type of inference *causal strategic inference.*

## A.2   Causal Strategic Inference

As mentioned above, our goal is to study interventions, the second type of causal queries, in game-theoretic settings. Since interventions are performed using surgeries, we now explore different types of surgeries that can be done in a game-theoretic setting. First, recall from the main body of the paper that a game in non-cooperative game theory can be described by a set of *players*, a set of *actions* for each player, and a *payoff function* for each player that maps each *joint-action* to a real number [44]. Here, a joint-action is specified by an action for each player. A central solution concept in non-cooperative game theory is Nash equilibrium. A pure-strategy Nash equilibrium (PSNE) can be defined as a joint-action of the players such that every player *plays* its *best response* to the other players' actions simultaneously. Here, a best-response of a player to the the other players' actions is defined by an action that maximizes its payoff with respect to the corresponding joint-action.

Although game theory explicitly represents the actions of the players, it is different from actions (e.g., the *do* operation) or interventions in the context of causal probabilistic inference. In game theory, actions are adopted or played by the players of the game, who are integral parts of the system. However, in causal probabilistic inference, interventions are performed by someone outside the system, such as an experiment designer. Notably, an intervention involving a set of random variables can be performed if one has control over them. One implication of such an intervention is that it makes *changes* to the original system (and hence the name surgery). Therefore, causal probabilistic inference is a query concerning the *changed system*, although the given input is with respect to the *system before the changes were made*. We will formulate causal strategic inference in an analogous way, but first, we will answer the question regarding the types of surgeries we perform. We will consider the following two possibilities.

**Surgery 1: Setting the Actions of Some of the Players**

One way of doing a surgery on a game is to restrict the actions of some of the players. For example, we can set the actions of some of the players to particular ones. Now, the question is: how should we interpret this surgery? For example, suppose that player $i$'s action has been set to $a_i$. Should we modify the game in a way that player $i$'s best response is always $a_i$, no matter what the other players play? Or, should we keep the game unchanged and rather focus only on those equilibria (if any) where player $i$ plays $a_i$ as its best response? Let us consider these two different interpretations.

First, once we set the actions of some of the players, we can modify the original game in the following way. Consider any player $i$ whose action has been fixed to $a_i$. The payoff function of player $i$ is changed so that $i$'s payoff is 1 for any joint-action in which $i$ plays $a_i$ and 0 for all other joint-actions. This change makes sure that the preset actions are indeed best responses of the corresponding players in the modified game (with respect to any action that the other players play). Note that a Nash equilibrium of the modified game may not be a Nash equilibrium of the original game. In particular, the players whose actions have been set may not be playing their best response with respect to the original game. Therefore, the outcome of the modified game is not guaranteed to be stable with respect to the original game. Such an approach has been used in a line of work on finding the most influential nodes in a social network [35], where the goal is to maximize the spread of a "new" behavior (e.g., buying a new product) by selecting a small "seed" set of initial adopters. The underlying mechanism is to set the actions of the seed players to the one denoting the adoption of the new behavior and then let the diffusion process set off. At the end of the diffusion process, however, the seed players may not be playing their best response *with respect to the original game.*

Controlling the actions of some of the players has also been used in other settings. For example, in the setting of network routing games, Sharma and Williamson study the minimum number of users that need to be controlled by a central authority to improve the social welfare of a Nash equilibrium [57]. This is motivated by the case where there are two types of users of a network application: premium users with the privilege of choosing their own route of traffic and ordinary users, who must go by whatever the network administrator has chosen for them [52]. The general problem is to find the minimum fraction of users to be controlled by the network administrator to achieve a desirable objective. Here, being controlled by the network administrator, the ordinary users might be forced to adopt an action that

they would not have adopted otherwise.

Second and in contrast to the above point, we can also do interventions by "controlling" the actions of some of the players *without changing the game*. For example, after setting the actions of some of the players, we can ask questions regarding the stable outcomes (e.g., Nash equilibria) where these players play according to the preset actions. An example of an inference question in this approach is to ask how many stable outcomes could possibly result from setting the actions of a subset of the players.

## Surgery 2: Changing the Structure of the Game

In this type of surgery, we change the game without setting the action of any of the players. Note that the notion of "changing the game" is very much open ended. It can potentially mean changing the payoff function of a player, removing a player from the game, adding a new player to the game, changing the set of actions of a player, as well as any combination of these. Here, we narrow our focus to the *structure* of a special type of games, namely graphical games in parametric forms (as opposed to normal forms). We use the term structure in this context to refer to the underlying topology of the game. One example of an intervention by changing the structure of a game is to remove a player from the game. A causal strategic inference question under this type of intervention is to infer how the outcome of the game would change due to the intervention.

We study such interventions in this paper, where we model a microfinance market in a game-theoretic way in order to ask causal strategic inference questions, such as what would happen if some of the loss-making government-owned banks are shut down? To answer such a question, we first learn the parameters of the model from real-world data and compute an equilibrium point, which reflects the outcome *before* the removal of any bank. We then do an intervention by changing the structure of the game (i.e., removing the loss-making government-owned banks). After that, we compute an equilibrium point of the resulting game (note that after removing the banks, we *do not* go back and learn the parameters of the model again). The difference in equilibrium outcomes before and after the removal of the banks gives us the desired answer.

# B   Comparison and Contrast with Econometrics Literature

We should first clarify that the idea of interventions in strategic settings is not new. Some of the surgeries we have mentioned above have been studied before in the context of various application scenarios. However, our main objective here is to build a comprehensive framework that wraps around this idea of interventions in strategic settings.

## B.1   Causal Strategic Inference in Econometrics

Modeling strategic scenarios and estimating the parameters of the model are active research topics in econometrics. Here, we will conduct a brief review of the literature with the goal of illustrating the difference between our approach and the general approach in econometrics. Since we will not give any detailed specification of our model, the discussion will be at a high level.

In econometrics literature, the flagship application scenario for studying strategic decision-making is the setting where two or more firms simultaneously decide whether to enter a market or not. This decision is strategic, because a firm's decision and hence its expected profit depend on the decisions of the other firms. As a result, game theory has been the prescriptive tool for modeling entry decisions in econometrics.

Within the general game-theoretic framework, there is a variety of entry-decision models in econometrics capturing homogeneous versus heterogeneous firms, complete information versus private information settings, and static versus dynamic games. The literature also shows different ways of addressing some of the inherent issues like the multiplicity of equilibria and equilibrium selection. There is, however, one unifying theme in the literature: almost all of the models are based on the discrete choice model [39, 40]. The discrete choice model in its originality does not allow the utility of an entity to depend on the actions of the other entities. However, the main ingredient of a game-theoretic model is this interdependence of actions. To account for this, the econometrics models that we will review indeed extend the original discrete choice model to what the literature commonly refers to as the *discrete game model*.

### B.1.1 Bjorn and Vuong's Model of Labor Force Participation

The first discrete game model is attributed to Bjorn and Vuong, who studied the case of simultaneous decision-making by a husband and a wife on whether to enter the labor force or not [12]. This is a two-player game, where each player has two actions. Denoting the action of player $i \in \{1, 2\}$ by $x_i \in \{0, 1\}$ (0 denotes not entering and 1 entering the labor force) and that of the other player by $x_{-i}$, the payoff of $i$ is defined using McFadden's random utility model [39] as follows.

$$\widetilde{u}_i(x_i, x_{-i}) = u_i(x_i, x_{-i}) + \eta_i(x_i, x_{-i}). \tag{1}$$

The above payoff function consists of an observed, deterministic part $u_i$ and an unobserved (to the researcher), random part $\eta_i$. The first accounts for observed attributes of the players, such as age, education level, assets, number of kids, etc. The latter part accounts for factors that the researcher could not observe or did not model, and as a result, it appears as a random shock. Every player $i$ observes its own random part $\eta_i$, but depending on whether it also observes the other player's random part, the game becomes either a complete information or an incomplete information game, respectively. The model of Bjorn and Vuong is a complete information one. Here, the best response $x_i^*$ of player $i$ can be written as follows.

$$x_i^* = 1 \iff \widetilde{u}_i(1, x_{-i}) - \widetilde{u}_i(0, x_{-i}) > 0. \tag{2}$$

In other words, player $i$'s best response is to choose the action that maximizes its payoff with respect to the other player's action. A pure-strategy Nash equilibrium (PSNE) is given by $(x_1^*, x_2^*)$ such that both of players are best responding to each other simultaneously. [1] Obviously, there could be three possible types of outcomes in this game: a unique PSNE, multiple PSNE, and no PSNE at all. Bjorn and Vuong view the data as a unique equilibrium. However, if the latter two possibilities of multiplicity and non-existence are ruled out, then they show that the model no longer remains strategic (that is, the best response of one of the two players does not depend on the other player's action). It rather becomes equivalent to a

previously studied *simultaneous equations model with structural shift* where a certain "logical consistency condition" must hold [29, 55]. Now, if the option of the multiplicity of PSNE is kept on table and if the data is viewed as a unique PSNE, an important question is: which one of the multiple possible PSNE is "played" in the data? This is typically known as the *equilibrium selection problem.*

Bjorn and Vuong take a randomized approach to this problem of equilibrium selection. They assign probabilities to all possible pairs of the reaction functions [2] of the two players and express the probability of each PSNE in terms of these probabilities. They then use the maximum likelihood technique to estimate the probability of observing any particular PSNE, along with the other parameters of the model that are not detailed here. They also give a necessary and sufficient condition for these parameters to be *identifiable*, which means that if that condition holds, then for any outcome of the model, the estimated parameters are unique. In other words, different instantiations of the parameters cannot generate the same outcome. In general, identifiability of the parameters is a major focus in econometrics literature.

Being the first of its kind, Bjorn and Vuong's model is simplistic and does not scale well if the number of players is increased. For instance, if there is a large number of players, then assigning a probability to each possible combination of the reaction functions of all the players would be computationally expensive. As we will see, much of the later literature actually avoids combining the reaction functions of the players.

### B.1.2 Entry Models of Bresnahan and Reiss

Bresnahan and Reiss investigate entry in monopoly markets using two types of models: a simultaneous, game-theoretic model and a sequential decision making model [14]. Their empirical study is based on the markets of automobile dealers with a focus on how market sizes influence entry decisions and whether the second entrant faces entry barrier (i.e., whether the fixed cost and market opportunities for the second entrant are less favorable compared to the first entrant).

This is again a two-player, two-action setting. We will first give a brief overview of the simultaneous-move model of Bresnahan and Reiss. Suppose that $\widetilde{u}_i^M$ and $\widetilde{u}_i^D$ are the payoffs of firm $i$ in a monopoly and a duopoly

market, respectively. We will not go into the details of these payoff functions, but as in the Bjorn-Vuong model, they also comprise of two parts: an observable part and an unobserved, random part. The random part is observed by both the players, although not by the researcher. The entry decision $(x_1^*, x_2^*)$ would be a pure-strategy Nash equilibrium (PSNE) if and only if the following best response condition holds for all firms $i = 1, 2$.

$$x_i^* = 1 \iff (1 - x_{-i}^*)\widetilde{u}_i^M + x_{-i}^*\widetilde{u}_i^D > 0.$$

Given a particular model, a PSNE outcome could be one of the following five types: monopoly by firm 1, monopoly by firm 2, duopoly, no entrant, and finally, monopoly by either firm 1 or firm 2 (but not duopoly). Once again, the multiplicity of PSNE is deemed challenging, as the authors say, "The presence of non-unique equilibria in game-theoretic models makes it impossible to use standard qualitative choice models to model entrants' profits." This is so, because the data is viewed as a unique PSNE. As Bjorn and Vuong showed earlier, restricting the model to rule out the multiplicity of PSNE results in a model that is not strategic any more. Therefore, the equilibrium selection problem arises inevitably.

Regarding the problem of equilibrium selection, Bresnahan and Reiss observe that if the model is reinterpreted to predict the *number of entrants* instead of the identity of the entrants, then the PSNE outcome is always unique. Another approach to avoid the multiplicity of PSNE is to consider a sequential-move version of the model. It is easy to show that if the firms do not make their decisions simultaneously, then the outcome is unique. The estimation of the model parameters using spatially isolated rural automobile dealership markets shows that the second entrant is not subjected to entry barrier and that its entry does not cause the first entrant's profits by much. In many cases, this is due to the market size being already very big when the second firm enters the market.

In a related paper, Bresnahan and Reiss discuss the issues of the existence and the uniqueness of PSNE in discrete game models [15]. They also discuss how one could deal with mixed-strategy Nash equilibria (MSNE) in discrete game models and how these models can be extended to the cooperative games setting. Although they motivated the issues of MSNE and cooperative games using real world examples, they did not actually apply their ideas to any empirical setting. For instance, they say that "the researcher must exercise care when selecting [certain probability] distributions," which needs to be done on a case by case basis if we would like to consider MSNE.

### B.1.3 Berry's Model of Entry in Airline Markets

Airline markets, each consisting of a source-destination pair of cities, have been studied by economists from different points of view. A common example is various explanations of an airline's profit due to its hub and spoke network [37]. Berry took a different approach to studying an airline market, by investigating the effects of strategic entry decisions of an airline on the profitability of the flights in a market [11]. To model the entry decision of an airline in a market, Berry presents a discrete game model that allows for a *large number of heterogeneous* airlines. Apart from the specifics of the model, this is one of the key differences with the previous entry models, such as the ones by Bresnahan and Reiss. Heterogeneity among the airlines can be observed in terms of their flight networks, fleets of aircrafts, etc. Heterogeneity can also be due to unobserved factors. In Berry's model, the payoff of an airline $i$ in a market $k$ is defined as follows.

$$u_{ik}(\mathbf{x}_k) = v_k(f(\mathbf{x}_k)) + \phi_{ik}.$$

Here, $\mathbf{x}_k$ is the vector of the entry decisions of each airline $i$ in a market $k$, $x_{ik} \in 0,1$ denotes an airlines entry decision. The first term in the payoff function is market specific and captures the competitive effect due to the entry decisions of the airlines, and the second term is specific to the airline-market pair and is treated as a single index of profitability. In order to guarantee the existence of a PSNE and the uniqueness of the *number* of PSNE (to deal with the equilibrium selection problem as mentioned above), Berry imposes several assumptions. First, the airlines in a market $k$ can be sorted according to the profitability index $\phi_{ik}$, and this ordering is independent of the entry decisions. Second, the function $f(\mathbf{x}_k)$ in the first term is defined as the count of entrants, i.e., $f(\mathbf{x}_k) = \sum_i x_{ik}$. Third, the market-specific function $v_k$ is decreasing in the number of entrants.

As before, each of the two terms in the above payoff function is further decomposed into two parts: one observable part and one unobserved, random part. Again, the setting here is a complete information game. The main challenge in analytically characterizing the probability of a certain number of entrants is due to the large number of airlines, which contributes to an exponential number of integrations over the random parts. Berry proposes two directions to address this. The first one is to impose additional restrictions on the model, such as removing the part of strategic interaction from the payoff function (i.e., an airline's profit is not affected by the number of entrants). The other direction is to apply simulation estimators [41]. The

estimated model shows a strong negative influence of competition on an airline's profit, which can limit the effectiveness of a policy encouraging the potential entrants.

As Berry points out, his entry model is guided by a "partial equilibrium approach," where instead of considering an airline's network of flight routes, the analysis focuses on a pair of source-destination cities. However, we know that the network structure of an airline's flight routes is one of the most important ingredients of its operation and profitability. In our view, the major challenge in accounting for this network structure is due to the analytical approach to the problem. An alternative to deal with this would be an *algorithmic approach*, which we pursue in this paper (although not on the same problem).

Berry's model has been subsequently extended by others. Ciliberto and Tamer allow a general form of heterogeneity among the airlines that no longer guarantees a unique number of entrants in all the PSNE of the (complete information) game [17]. Without assuming a particular equilibrium selection rule, they *bound* the choice probabilities between an upper and a lower limit. They estimate the parameters of the model by minimizing the distance between the *set* of choice probabilities between these two bounds and the probabilities estimated from the data. Apart from modeling the airline industry, a different work by Berry et al. presents techniques to estimate the parameters of an oligopolistic market with a wide range of product differentiation with applications to the U.S. automobile market [10].

### B.1.4   Other Models

The three early models that we reviewed above exhibit some of the key aspects of the general econometrics approach to modeling strategic scenarios, such as the adoption of a random utility model [39, 40], analytical characterizations of some of the quantities of interest, a way of dealing with the multiplicity of PSNE (for example, a randomized equilibrium selection mechanism) or a way of avoiding the multiplicity issue altogether (for example, by imposing additional assumptions that would lead to a unique equilibrium or by reinterpreting the model to predict a common property of all PSNE, such as the number of players playing a particular action), and the identifiability of the parameters of the model. The literature has since been enriched with a number of interesting pieces of work extending the previous research as well as injecting new ideas to deal with these challenging tasks. Here, we will briefly review a sample of some of the widely cited research along this line.

### B.1.5    Seim's Model of Product Differentiation

The early game-theoretic entry models, such as the one by Berry [11], focus mostly on firm-specific profits and the competitive effects of multiple entrants, but do not model product differentiation by the firms. Seim proposes a discrete game model of entry decisions that allows the firms to spatially differentiate their products by choosing, for example, a location of operation [56]. Her empirical study is based on the location choice of video retailers. In contrast to much of the earlier work, she models entry decisions as a game of *incomplete* information, which accounts for a firm's lack of information about many of the characteristics of another firm, such as that firm's managerial talent. Interestingly, the reason why many of the earlier models were games of *complete* information is that the incomplete information version was thought to be more challenging. For example, Bresnahan and Reiss say, "Games of private information pose much more complicated estimation issues" [15, p. 60]. However, it later turned out that an equilibrium in an incomplete information game (also known as the Bayes-Nash equilibrium) can be characterized more easily [53] and that the estimation can also be done in a straightforward two-step method [7]. As a result, modeling a scenario as an incomplete information game often serves the dual purpose of modeling various unobserved idiosyncrasies as well as dealing with that model in a tractable fashion.

Going back to the model of Seim, a market consists of multiple locations, and each firm chooses an entry location if it decides to enter the market. The choices of the firms are made simultaneously. The payoff function of a firm consists of the following terms: an observable location-specific characteristics (such as the population and the income level of the potential customers), an unobserved market-specific random term, a competitive effect term that accounts for the decisions of all the firms, and an unobserved firm and location-specific term that captures the firm's private information about its profitability in that location. The last term is assumed to be independently and identically distributed (iid) draws from a type-1 extreme value distribution, which leads to closed form expressions for a firm's choice probabilities. The goal of the model is to predict the unique number of entrants in a PSNE, although Seim also shows that the PSNE itself is unique under certain additional assumptions. The estimated model shows that video retailers use location choice to their competitive advantage. Also, as the market size increases, the local demand decreases due to the spreading out of the population density. As a result, the number of entrants does not increase by much.

### B.1.6   Augereau et al.'s Model of Technology Adoption

Beyond entry decisions, discrete game models have been designed for many
other interesting phenomena, such as adoption of a particular technology.
Augereau et al. study the adoption of the 56K modem technology by the
ISPs in a market during the late 1990s [3]. At that time, there were two
competing and incompatible implementations of the 56K standard, one by
the U.S. Robotics and the other by Rockwell. If an ISP adopts the U.S.
Robotics technology, for example, then its customers must also buy the
U.S. Robotics modems to enjoy a high-speed connection. Augereau et al.
model the choice of the ISPs as a discrete game of incomplete information,
which accounts for the characteristics of the market as well as the ISPs and
the simultaneous decision of the ISPs. They show that the ISPs in a market
want their choice to be different from their competitors (so that they do not
lose their customers to their competitors).

### B.1.7   Sweeting's Model of the Timing of Radio Commercials

Whereas Augereau et al.'s result on technology adoption among ISPs can
be interpreted as a coordination failure, Sweeting's model of radio stations'
choice of timings for playing commercials tells us the opposite story of co-
ordination and synchronization. In Sweeting's model, the radio stations in
a market choose timings for advertisement strategically with their payoff
function capturing market and station specific factors, a competitive factor
that takes the form of the proportion of the other stations that choose the
same time, and a private information term modeled as a random shock.
Sweeting shows how the multiplicity of equilibria can help the identification
of the parameters of the model. Estimation is done by the two-step method
of Bajari et al. mentioned above [7]. The finding of coordination among the
radio stations signifies that the interests of the radio stations are somewhat
aligned with the interests of the advertisers. During drivetime hours, the
coordination incentive is very strong. Multiplicity of equilibria is also more
common during that time.

### B.1.8   Discrete Game Models of the Banking Sector: ATM Net-
###          works

Game-theoretic models have also been developed to capture decision mak-
ing in the banking sector. Consider the case of ATM networks for example.
From our daily experience, the ATM networks of different banks are incom-
patible unless we pay a surcharge. This surcharge never covers the cost of

the ATM service of a bank. It is rather intended to attract customers to open deposit accounts at the bank. As a result, larger banks often charge more surcharge than smaller banks and credit unions.

Ishii models banks and customers as strategic decision makers in a two-sided market to understand the effects of this surcharge [31]. In brief, the payoff function of a customer for choosing a bank (i.e., having a deposit account in that bank) in a market accounts for the customer's observable characteristics, the bank's observable characteristics (e.g., its number of ATMs), the bank's interest rate (which is determined by the bank strategically), and the unobservables corresponding to the customer and the bank. The banks, on the other hand, maximize their profits in two stages. In the first stage, each bank strategically chooses the number of ATMs to be deployed. In the second stage, it chooses an interest rate to maximize its profit given a PSNE from the previous stage. We will not go into the details of these two stages. The estimation is done by the generalized method of moments (GMM) method. The estimated model captures the phenomenon that when choosing a bank, customers are influenced by the bank's ATM network size and its surcharge. It also shows that the revenue from the ATM service does not cover its cost. Rather, the incentive for a bank to invest in an ATM network lies in securing a share of the deposit market.

An interesting feature of Ishii's work is her study of various counterfactuals. For example, what would happen if the surcharges are eliminated by law? In that case, the model predicts that the market becomes less concentrated. That is, the market share of customer deposits is reallocated from the larger banks to the smaller ones. In response, the larger banks raise the interest rates on deposit, which decreases their profit. In fact, the overall profit of the industry decreases due to the elimination of surcharges. To the contrary, the presence of surcharges encourages banks to expand their ATM networks, although it makes the market share of customer deposits concentrated at the larger banks.

### B.1.9 Discrete Game Models of the Banking Sector: Adoption of ACH

Game-theoretic models of decision making have also been developed to study other phenomena in the banking industry, such as the adoption of automated clearinghouse (ACH) technology, which provides an electronic equivalent of paper checks and is commonly used in direct deposits and automated bill payments. Ackerberg and Gowrisankaran estimate the magnitude of network effects in ACH adoption [1]. The model has two sides: banks and

customers (e.g., small businesses), where the customers are treated as homogeneous. Two banks can do an ACH transaction if both have already adopted this technology. On the other hand, there are two alternatives for a customer. In a one-way transaction, a customer may receive ACH payment without adopting the technology, provided that her bank has adopted it. In a two-way transaction, a customer must also adopt the ACH technology in addition to her bank.

Ackerberg and Gowrisankaran define a two-stage game. In the first stage, the banks simultaneously decide whether to adopt ACH or not, and in the second stage, the customers decide on the adoption of ACH, given the decision of the banks. This model does not rule out the multiplicity of equilibria. The way the authors approach equilibrium selection is by estimating the probability of seeing one of the two extreme equilibria (Pareto-best and Pareto-worst), which is obviously a simplification compared to considering all possible equilibria. The estimation of the parameters is done by a simulation method. The estimated model is used for counterfactual policy experiments. The model suggests that government subsidy directed toward the customers is more effective for ACH adoption than that directed toward the banks.

### B.1.10   Recent Developments

The subject of discrete game models is an active area of research in econometrics. Recently, Bajari, Hong, and Ryan presented methods for identification and estimation of discrete game models of complete information, which are also applicable to general normal-form games [8]. A key feature of their work is that they estimate both the parameters of the model and the equilibrium selection mechanism. For the latter, they compute all the equilibria of a game using the algorithm of McKelvey and McLennan [42]. This is certainly a computationally expensive step for large games. In a separate work, Bajari et al. address estimation of discrete game models of incomplete information [6]. Finally, an excellent survey of some of the recent results on the variants of discrete game models (e.g., complete vs. incomplete information, static vs. dynamic games) and their identification, estimation, and equilibrium selection has been presented by Bajar, Hong, and Nekipelov [7].

## B.2 Causal Strategic Inference: Our Approach

Even though we study a completely different set of problems than the ones reviewed above, we share the most important ingredient of strategic decision making with all these problem. However, there are some fundamental differences between our approach to studying strategic settings and that of econometricians in general. Once again, since we have not formally defined any of our models, this discussion will not focus on any detail. It will rather highlight these key differences in the context of causal strategic inference.

### B.2.1 Analytic vs. Algorithmic Approaches

First, an econometrics approach to dealing with strategic settings is in large part analytic. True, econometricians do provide algorithms (e.g., algorithms for estimating parameters), but for the most part, those algorithms are primarily driven by analytic techniques. See, for example, the two-step estimator for discrete games of incomplete information [7]. In contrast, the main focus of this paper is an algorithmic approach to problems.

Note that we do not claim that an algorithmic approach is better than an analytic approach. However, in certain situations, an algorithmic approach might provide a good alternative to an analytic approach. This is particularly the case when we have large, complex systems with an underlying structure. For example, the network structures of an airline's flight routes are not often exploited in the econometrics models of airline markets [11]. This could be due to the challenge posed by dealing with a large, heterogeneous system in an analytic manner. As mentioned above, such complex systems are exactly the focus of this paper. We show that looking through an algorithmic lens helps us solve problems that would otherwise be impossible to manage analytically.

### B.2.2 Modeling

A common modeling approach in the econometrics literature that we reviewed is to adopt of the random utility model [39, 40]. Besides giving a reasonable way of modeling unobservables, this approach sometimes also leads to simple closed form expressions that are easy to deal with. For example, the choice probabilities in a discrete game model can be expressed that way when the random parts of the payoff functions are iid type-1 extreme value distributed.

In contrast, the modeling approach in this paper is completely different. We model two-sided microfinance markets using the well-studied concept of

abstract economies in classical economics [19, 2]. Our model of microfinance markets is network-structured, and a very special case of it can be shown to be one type of Fisher market [23], which has been a subject of intense algorithmic study by computer scientists in recent years.

Again, we do not claim that our models are "superior" to the random utility model in any sense. Rather, with the specific applications that we would like to address here, our modeling approach serves the purpose best while having its root in the relevant social science literature. The key aspects of our models are heterogeneity, network-structure, compact representation, and the ability to capture the strategic interactions among a large number of entities.

### B.2.3   Estimation

Econometrics and computer science (machine learning, in particular) have diverging views on the issue of the estimation of parameters. As we saw in the literature review above, the identification of the parameters of a model (or some function of the parameters) is a major concern in econometrics literature. The reason is that if the parameters are not identified (i.e., different instantiations of the parameters lead to the same outcome), then the estimated model may be very different from the actual system that generates the data, even though the estimated model produces almost the same outcome as the actual system. To ensure identification, additional restrictions are sometimes imposed on the model. Identification with infinite sample is also very common [8, 6].

In contrast, one of the most primitive principles of machine learning is Occam's razor, which says that if multiple models explain the same observation reasonably well, then we should choose the "simplest" model. Compared to a complex model, a simpler model usually also shows more generalization power in terms of predicting something that has not been observed before. However, too much simplicity might not capture the observed data well. Therefore, researchers often strike a balance between simplicity and complexity, which is usually guided by the well-known bias-variance tradeoff (a high bias corresponds to too much simplicity and a high variance corresponds to complexity). One technique often employed for this is to estimate the parameters of a model using a portion of the data, not the whole, and then to test its predictive power using the rest of the data. At the end, the best predictive model is chosen.

Estimation in econometrics, on the other hand, is very different from that in machine learning. In econometrics, all the available data is used

for the purpose of estimation, and a high variance is a desirable objective. The anxiety about whether the estimated model would perform well in an unforeseen environment is eased with the assurance that the parameters have been identified. However, as we mentioned above, identification often necessitates making strong assumptions.

There are many other contested issues between these two disciplines, which are out of scope for this paper. It is not that one of the approaches is good and the other is bad. It is just that they are different. In this paper, we have taken a machine learning approach to estimation. The estimation is done using a bi-level optimization program. In our work, the predictive power of a model with respect to unforeseen events has been the prime focus.

The objective of our estimation is also different from that of the fast growing literature on causal estimation in computer science and statistics. For example, a common technique for understanding the effects of new product features on consumers is known as bucket testing, which basically exposes the feature to a random sample of the population and measures its effect on them. With the advent of online social networks like Facebook, bucket testing can no longer focus on a disconnected random sample of users. It also needs to consider the network structure, because in the context of online social networks, the effect of a new feature is more meaningful when a user as well as some of her friends are exposed to it, compared to only the user being exposed in isolation.

To extend bucket testing to networked settings, Backstrom and Kleinberg propose a graph-theoretic sampling technique that addresses these two competing requirements: samples need to be uniformly random and they also need to be well-connected [5]. Along the same line, Ugander et al. propose graph cluster randomization techniques to give an efficient algorithm to compute the probability of the exposure of a user [59]. They also show that their techniques can lower the estimator variance. In another notable work, Toulis and Kao propose two techniques for estimating causal peer influence effects—a frequentist approach that can deal with more complex response functions and a Baysian approach that provides more accurate estimates under network uncertainties [58]. In contrast to this line of work, our goal is not causal estimation. We rather want to estimate models that capture strategic interactions.

### B.2.4 Equilibrium Selection

Almost all of the models we reviewed above exhibit multiplicity of equilibria. Therefore, the question of equilibrium selection naturally arises as

the data is often viewed as a single equilibrium. Econometrics literature suggests three main ways of dealing with the equilibrium selection problem [8, 7]. First, the probability that an equilibrium, which is generated by the model, is observed in the data is estimated [12, 8]. Sometimes, instead of considering all possible equilibria, only a few equilibria are considered in this probabilistic approach [1]. Second, the model can sometimes be reinterpreted to give a unique outcome, even if there are multiple equilibria. A typical example is considering the number of entrants instead of the identity of entrants in an equilibrium of an entry market [14, 11]. Third, the choice probabilities can sometimes be bounded between two limits, which guides the selection of an equilibrium [17].

In our study of microfinance markets, our model can potentially generate multiple equilibria. We select one of these equilibria that is geometrically closest to the observed data. This equilibrium selection mechanism is embedded in the parameter estimation procedure. We also test for the robustness of this mechanism. We find that even if we introduce considerably large magnitudes of noise in the data, this mechanism selects the same equilibrium.

### B.2.5 Interventions

A key component of causal strategic inference as well as causality in general is interventions. All of the econometrics studies reviewed above concern two of the components of causal strategic inference that we mentioned earlier: modeling a strategic scenario and estimating the parameters of the model. However, many of these studies do not perform interventions. There are, of course, exceptions. For example, Isii studies the effect of removing the ATM surcharges [31]. Ackerberg and Gowrisankaran show the comparative effects of subsides to customers and banks on ACH adoption [1].

The main focus of this work is a wide range of interventions. We perform various interventions in a microfinance market, such as setting an interest rate cap, removing a bank from the system, providing subsidies to certain banks to make loans more affordable, etc. It should be mentioned here that interventions by removing players is not a new concept. Ballester et al., for example, performed interventions in a criminal network by removing players from it [9].

## B.3   Connection to Existing Models of Networked Economies

Our model is essentially one of networked economy. In recent times, there has been an intense, inter-disciplinary research effort in the area of networked economies, primarily undertaken by the economics and the computer science communities and in many instances jointly by researchers from these two communities. Subjects of investigation have ranged from generalizing abstract economies in a graphical setting [33], modeling networked markets, such as labor markets and trades (see, for example, Chapter 10 of [32]), designing mechanisms with desirable properties for such markets [4], analyzing how properties such as competition [13] and price variation [34] are influenced by the underlying network structure, to the most fundamental algorithmic question of computing an equilibrium point in such settings [54, 33]. Although we postpone a formal description of our model, we will now place our model in the context of the existing ones at a very high level.

Let us begin with the Fisher model [23], which consists of a set of buyers and a set of divisible goods sold by one central seller (i.e., a fully connected network). The buyers come to the market with some initial endowments of money, and each has a utility function over bundles of goods. Given the prices of the goods, their objective is to use their endowment to purchase a bundle of goods that maximizes their utility. An equilibrium point consists of the unit prices and the allocations of goods such that each buyer fulfills his objective and in addition, there is no excess demand or excess supply (i.e., the market clears). A *graphical* Fisher model with one good [34] consists of a set of buyers and *a set of sellers*. All the sellers sell the same good, but the important aspect of this model is that each buyer has access to a subset of the sellers, not necessarily all the sellers. An equilibrium point in this graphical setting is defined similar to the original one. An important distinction between our model and graphical Fisher model is that our model allows buyers (i.e., villages) to invest the goods (i.e., loans) in productive projects, thereby generating revenue that can be used to pay for the goods (i.e., repay the loans). In other words, the crucial modeling parameter of "endowment" is no longer a constant in our case. Furthermore, in our model, the villages have a very different objective function than the one in a Fisher model [34, 60]. There is, however, an interesting connection between our model and that of Fisher through an Eisenberg-Gale convex program formulation, which we showed in the main body of the paper.

Arrow and Debreu gave a very generalized model of economy in their seminal work on competitive economies [2]. In fact, a Fisher economy is a special case of an Arrow-Debreu economy (see, for example, [60]). Arrow

and Debreu's proof of the existence of an equilibrium point uses Debreu's concept of abstract economy [19], which interestingly generalizes Nash's non-cooperative games [43] in the following way. In an abstract economy, not only a player's payoff but also her domain of actions are affected by another player's choice of action. Putting our model in the context of an abstract economy, a village's set of possible demands for loan from an MFI depends on the MFI's interest rate. For example, in the simplest setting of one MFI operating in one village, the village cannot ask for unlimited amount of loan from the MFI if the MFI's interest rate is above a certain bound. However, for the same reasons cited in the previous paragraph (i.e., variable endowment), the classical Arrow-Debreu model or more specifically, a recently developed graphical extension to the Arrow-Debreu model [33], does not capture our setting.

Our work is different from various other works on networked economies [36, 4, 54, 26, 22, 13] from the perspectives of modeling, problem specification, and application. For example, Kranton and Minehart model buyer-seller exchange economies as networks with an emphasis on the emergence of links in such networks [36]. They show that although buyers and sellers are modeled as self-interested non-cooperative agents, "efficient" network structures are necessarily equilibrium outcomes and that for a restricted case, equilibrium outcomes are necessarily efficient. In a related work of significant implications, Even-Dar *et al.* completely characterize the set of all buyer-seller network structures that are equilibrium outcomes in their model of exchange economies [22]. In contrast to these works, we do not study network formation here, i.e., we treat the spatial structure of the branch-banking MFIs as exogenous.

# C    Complete Proofs

We use the following two optimization programs throughout this section.
    *MFI-side optimization problem.*

$$\max_{r_i} \quad 1$$

$$\text{subject to} \quad r_i \left( T_i - \sum_{j \in V_i} x_{j,i} \right) = 0$$

$$\sum_{j \in V_i} x_{j,i} \leq T_i \qquad (P_M)$$

$$r_i \geq 0$$

*Village-side optimization problem.*

$$\max_{\mathbf{x}_j = (x_{j,i})_{i \in B_j}} \quad \sum_{i \in B_j} x_{j,i} + \lambda \sum_{i \in B_j} x_{j,i} \log \frac{1}{x_{j,i}}$$

$$\text{subject to} \quad \sum_{i \in B_j} x_{j,i}(1 + r_i - e_j) \leq d_j \qquad (P_V)$$

$$\mathbf{x}_j \geq 0$$

## C.1  Special Case: No Diversification of Loan Portfolios

**Property C.1.** *At any equilibrium point $(\mathbf{x}^*, \mathbf{r}^*)$, every MFI $i$'s supply must match the demand for its loan, i.e., $\sum_{j \in V_i} x_{j,i}^* = T_i$. Furthermore, every village $j$ borrows only from those MFIs $i \in B_j$ that offer the lowest interest rate. That is, $\sum_{i \in B_j, r_i^* = r_{m_j}^*} x_{j,i}^*(1 + r_i^* - e_j) = d_j$ for $m_j \in \operatorname{argmin}_{i \in B_j} r_i^*$, and $x_{j,k}^* = 0$ for any MFI $k \notin m_j$.*

*Proof.* Suppose that there is an MFI $i$ such that at an equilibrium point we have $\sum_{j \in V_i} x_{j,i}^* < T_i$. Clearly, in this situation, MFI $i$'s constraint of $r_i^* \left( T_i - \sum_{j \in V_i} x_{j,i}^* \right) = 0$ can only be satisfied if $r_i^* = 0$. However, if $r_i^* = 0$ then for any village $j \in V_i$, the optimal demand $x_{j,i}^*$ will be unbounded. This happens because each village $j$ wants to maximize $\sum_{i \in B_j} x_{j,i}$, and with $r_i^* = 0$ and $e_j \geq 1$, the term $(1 + r_i^* - e_j)$ in the first constraint of $(P_V)$ becomes $\leq 0$. Since $d_j > 0$, that constraint is satisfied for $x_{j,i}^* = +\infty$. But $x_{j,i}^* = +\infty$ contradicts the constraint $\sum_{j \in V_i} x_{j,i}^* \leq T_i$ in the MFI side $(P_M)$. Therefore, for every MFI $i$, $\sum_{j \in V_i} x_{j,i}^* = T_i$ must hold at any equilibrium point. [3]

For the second claim, in order to maximize its objective function $\sum_{i \in B_j} x^*_{j,i}$, every village $j$ will be interested to borrow only from those MFIs that have the minimum interest rate $r^*_{m_j}$, where $m_j \in \arg\min_{i \in B_j} r^*_i$. Furthermore, at any equilibrium point, each village's budget constraint, i.e., the first constraint in $(P_V)$, must hold with equality. Otherwise, suppose that the following strict inequality holds for some village $j$ at an equilibrium point: $\sum_{i \in B_j, r^*_i = r^*_{m_j}} x^*_{j,i}(1 + r^*_i - e_j) < d_j$. Since this is a strict inequality, village $j$ can still increase its objective function $\sum_{i \in B_j} x^*_{j,i}$. Therefore, village $j$ is not maximizing its objective function, contradicting our assumption that this is an equilibrium point. $\qquad\square$

**Property C.2.** *At any equilibrium point* $(\mathbf{x}^*, \mathbf{r}^*)$, *for every MFI* $i$, $r^*_i > \max_{j \in V_i} e_j - 1$.

*Proof.* Consider any MFI $i$. Let $k \in \arg\max_{j \in V_i} e_j$. Suppose that $r^*_i \leq e_k - 1$. Then we get $(1 + r^*_i - e_k) \leq 0$. This allows $x^*_{j,i}$ to go to $+\infty$, and that violates the constraint $\sum_{j \in V_i} x^*_{j,i} \leq T_i$, contradicting the assumption that this is an equilibrium point. [4] $\qquad\square$

Following are two related results that preclude certain trivial allocations at an equilibrium point.

**Property C.3.** *At any equilibrium point* $(\mathbf{x}^*, \mathbf{r}^*)$, *for any village* $j$, *there exists an MFI* $i \in B_j$ *such that* $x^*_{j,i} > 0$.

*Proof.* Suppose that for some village $j$, and for all $i \in B_j$ we have $x^*_{j,i} = 0$. Since $d_j > 0$, the constraint $\sum_{i \in B_j} x^*_{j,i}(1 + r^*_i - e_j) \leq d_j$ of $(P_V)$ is satisfied, but village $j$ is not maximizing $\sum_{i \in B_j} x^*_{j,i}$. This contradicts that $(\mathbf{x}^*, \mathbf{r}^*)$ is an equilibrium point. $\qquad\square$

**Property C.4.** *At any equilibrium point* $(\mathbf{x}^*, \mathbf{r}^*)$, *for any MFI* $i$, *there exists a village* $j \in V_i$ *such that* $x^*_{j,i} > 0$.

---

to demand $x^*_{j,i} = +\infty$ even though $T_i$ is finite for MFI $i$. As mentioned earlier, the reason is that the MFI-side optimization problem $(P_M)$ treats $x^*_{j,i}$ as exogenous and does not have a direct control over it inside $(P_M)$. Moreover, the village-side optimization problem $(P_V)$ for village $j$ selects $(x^*_{j,i})_{i \in B_j}$ in order to maximize its objective function $\sum_{i \in B_j} x^*_{j,i}$, without considering the MFI-side constraint $\sum_{j \in V_i} x^*_{j,i} \leq T_i$. The contradiction is due to the necessary condition that *at any equilibrium point* $(\mathbf{x}^*, \mathbf{r}^*)$, all the constraints of both $(P_M)$ and $(P_V)$ must be satisfied.

[4]Once again, village $j$'s demand $x^*_{j,i}$ is determined in $(P_V)$ without trying to satisfy the constraints of $(P_M)$. However, at an equilibrium point $(\mathbf{x}^*, \mathbf{r}^*)$, both the village-side and the MFI-side problems must be optimized simultaneously.

*Proof.* If there exists an MFI $i$ such that for all villages $j \in V_i$, $x_{j,i}^* = 0$, then this violates the first constraint of $(P_M)$ in the following way. By Property C.2, $r_i^* > 0$, and by our modeling assumption, $T_i > 0$. Therefore, $r_i^* \left( T_i - \sum_{j \in V_i} x_{j,i}^* \right) > 0$. $\qquad\square$

## C.2 Eisenberg-Gale Formulation

We now present an Eisenberg-Gale convex program formulation of a restricted case of our model where the diversification parameter $\lambda = 0$ and all the villages $j$, $1 \leq j \leq m$, have the same revenue generation function $g_j(l) := d + el$, where $d > 0$ and $e \geq 1$ are constants. We will first prove that this case is equivalent to the following Eisenberg-Gale convex program [21, 60], which will give us the existence of an equilibrium point and the uniqueness of the equilibrium interest rates as a corollary.

Let us explain our overall plan here. We will first write down the Eisenberg-Gale convex program $(P_E)$ below. We will then make a connection between an equilibrium point $(\mathbf{x}^*, \mathbf{r}^*)$ of a microfinance market and the variables of program $(P_E)$. In particular, we will define $x_{j,i}^* \equiv z_{j,i}^*$ and express $r_i^*$ in terms of certain dual variables of $(P_E)$. Once we do that, we will show that the equilibrium conditions of $(P_M)$ and $(P_V)$ for the above mentioned special case are equivalent to the Karush-Kuhn-Tucker (KKT) conditions of $(P_E)$. Let us begin by writing down the Eisenberg-Gale program $(P_E)$. [5]

$$
\begin{aligned}
\min_{\mathbf{z}} \quad & \sum_{j=1}^{m} -\log \sum_{i \in B_j} z_{j,i} \\
\text{subject to} \quad & \sum_{j \in V_i} z_{j,i} - T_i \leq 0, \quad 1 \leq i \leq n \qquad\qquad (P_E) \\
& z_{j,i} \geq 0, \qquad\qquad\quad 1 \leq i \leq n, \ j \in V_i
\end{aligned}
$$

Following are the Karush-Kuhn-Tucker (KKT) conditions for $(P_E)$.

*Stationary condition:*

$$
\nabla_z \left( \sum_{j=1}^{m} -\log \sum_{i \in B_j} z_{j,i} \right) + \sum_{i=1}^{n} \gamma_i \nabla_z \left( \sum_{j \in V_i} z_{j,i} - T_i \right) + \sum_{i=1}^{n} \sum_{j \in V_i} \mu_{j,i} \nabla_z \left( -z_{j,i} \right) = 0
$$

Evaluating this at $z_{j,i}^*$ for any $i \in \{1, ..., n\}$ and any $j \in V_i$, we obtain the following. [6]

$$-\frac{1}{\sum_{k \in B_j} z_{j,k}^*} + \gamma_i^* - \mu_{j,i}^* = 0 \tag{3}$$

*Primal feasibility:*

$$\sum_{j \in V_i} z_{j,i}^* - T_i \leq 0, \qquad 1 \leq i \leq n$$

$$z_{j,i}^* \geq 0, \qquad\qquad 1 \leq i \leq n \; j \in V_i$$

*Dual feasibility:*

$$\gamma_i^* \geq 0, \qquad 1 \leq i \leq n$$

$$\mu_{j,i}^* \geq 0, \qquad 1 \leq i \leq n, \; j \in V_i$$

*Complementary slackness:*

$$\gamma_i^* \left( \sum_{j \in V_i} z_{j,i}^* - T_i \right) = 0, \qquad 1 \leq i \leq n \tag{4}$$

$$\mu_{j,i}^* \left( -z_{j,i}^* \right) = 0, \qquad\qquad 1 \leq i \leq n, \; j \in V_i \tag{5}$$

Note that if $\gamma_i^* > 0$, then (4) gives us the following.

$$\sum_{j \in V_i} z_{j,i}^* - T_i = 0, \qquad 1 \leq i \leq n \tag{6}$$

Furthermore, if $z_{j,i}^* > 0$ then (5) implies $\mu_{j,i}^* = 0$. In that case, we obtain the following from the stationary condition (3).

$$\gamma_i^* = \frac{1}{\sum_{k \in B_j} z_{j,k}^*} \tag{7}$$

The following properties are obtained from the optimality condition of the above Eisenberg-Gale convex program $(P_E)$.

**Lemma C.5.** *For any $i$, there exists a $j \in V_i$ such that $z_{j,i}^* > 0$.*

*Proof.* Suppose that for some $i$, and for all $j \in V_i$, $z^*_{j,i} = 0$. This contradicts the optimality condition, because $T_i > 0$, and $\sum_{j \in V_i} z^*_{j,i} = 0$. Thus, $\sum_{j \in V_i} z^*_{j,i} - T_i < 0$, and $\sum_{j=1}^m - \log \sum_{i \in B_j} z^*_{j,i}$ can be further decreased by increasing the value of $z^*_{j,i}$ for some $j$. $\qquad \square$

Let us define $I^*(j) \equiv \{i \mid z^*_{j,i} > 0\}$. We will later see that this represents the set of MFIs from which a village $j$ borrows at an equilibrium point.

**Lemma C.6.** *For any $j$, $|I^*(j)| > 0$.*

*Proof.* Suppose that $z^*_{j,i} = 0$ for some $j$ and for all $i \in B_j$. Rearranging the terms of (3), we have for any $i \in B_j$:

$$\gamma^*_i = \frac{1}{\sum_{k \in B_j} z^*_{j,k}} + \mu^*_{j,i}.$$

Since $\mu^*_{j,i} \geq 0$ by the dual feasibility condition, we have $\gamma^*_i = +\infty$ from the above expression. This contradicts the complementary slackness condition (4), because $T_i > 0$ by our modeling assumption. Therefore, for any $j$ and some $i \in B_j$, $z^*_{j,i} > 0$, which completes the proof. $\qquad \square$

Another way of proving Lemma C.6 is to note that if $z^*_{j,i} = 0$ for some $j$ and for all $i \in B_j$, then the objective function of the Eisenberg-Gale program $(P_E)$ goes to $+\infty$. This cannot happen, because $(P_E)$ is minimizing the objective function, and the program is guaranteed to have a bounded optimal solution (for example, one bounded feasible solution is achieved by $z_{j,i} = \frac{T_i}{|V_i|}$ for all $i \in \{1, ..., n\}$ and all $j \in V_i$).

We can use Lemma C.6 to rewrite (7) in terms of $I^*(j)$. For any $j$ and any $i^*(j) \in I^*(j)$, the following holds.

$$\gamma^*_{i^*(j)} = \frac{1}{\sum_{k \in B_j} z^*_{j,k}} \tag{8}$$

To present an Eisenberg-Gale formulation of our market model given by $(P_M)$ and $(P_V)$, we define the following terms.

$$x^*_{j,i} \equiv z^*_{j,i}, \qquad \qquad \text{for all } i \in \{1, ..., n\} \text{ and all } j \in V_i \tag{9}$$
$$r^*_{i^*(j)} \equiv \gamma^*_{i^*(j)} d + e - 1, \qquad \text{for all } j \in \{1, ..., m\} \text{ and all } i^*(j) \in I^*(j) \tag{10}$$

Note that in (10) above, $r^*_i$ has not been explicitly defined for all $i \in 1, ..., n$. We first prove that $\cup_{j \in \{1,...,m\}} I^*(j) = \{1, ..., n\}$ (that is, for all $i$ we have defined $r^*_i$ above). We then prove that if $i^*(j) = i^*(j')$, where

$i^*(j) \in I^*(j)$ and $i^*(j') \in I^*(j')$ for $j \neq j'$, then $r^*_{i^*(j)} = r^*_{i^*(j')}$ (that is, if the same $i$ appears in two different $I^*(.)$, then the definition of $r^*_i$ is consistent with respect to these two cases).

For the first claim, suppose that for some $i$, $r^*_i$ has not been defined. This implies that for all $j$, $i \notin I^*(j)$. That is, for all $j$, $z^*_{j,i} = 0$, which violates Lemma C.5.

For the second claim, consider the definition of $r^*_{i^*(j)}$.

$$
\begin{aligned}
r^*_{i^*(j)} \quad &= \gamma^*_{i^*(j)} d + e - 1 \\
&= \gamma^*_{i^*(j')} d + e - 1 \qquad \text{[Since } i^*(j) = i^*(j')] \\
&= r^*_{i^*(j')} \qquad\qquad\qquad \text{[By definition]}
\end{aligned}
$$

Next, we use the definitions (9) and (10) above to show that none of the villages has any left-over money.

$$
\begin{aligned}
d \quad &= \frac{1 + r^*_{i^*(j)} - e}{\gamma^*_{i^*(j)}} \qquad\qquad \text{[Rearranging (10)]} \\
&= (1 + r^*_{i^*(j)} - e) \sum_{k \in B_j} z^*_{j,k} \qquad \text{[Using (8)]} \qquad\qquad (11)
\end{aligned}
$$

Next, we show that for any $i^*(j) \in I^*(j)$,

$$
\gamma^*_{i^*(j)} = \min_{k \in B_j} \gamma^*_k.
$$

By Equation (8), for any $i^*(j) \in I^*(j)$,

$$
\gamma^*_{i^*(j)} = \frac{1}{\sum_{k \in B_j} z^*_{j,k}}.
$$

For any $l \in B_j$ , we obtain from the stationary condition,

$$
\gamma^*_l \geq \frac{1}{\sum_{k \in B_j} z^*_{j,k}}, \text{ since } \mu^*_{j,l} \geq 0.
$$

Therefore, for any $i^*(j) \in I^*(j)$,

$$
\gamma^*_{i^*(j)} = \min_{k \in B_j} \gamma^*_k.
$$

Thus, using the definition of $r$, $r^*_{i^*(j)} = \min_{k \in B_j} r^*_k$. Furthermore, for any $l \in B_j - I^*(j)$, $z^*_{j,l} = 0$. We obtain from (11),

$$
\begin{aligned}
d &= (1 + \min_{l \in B_j} r^*_l - e) \sum_{k \in B_j} z^*_{j,k} \\
&= \sum_{k \in B_j} z^*_{j,k}(1 + r^*_k - e).
\end{aligned}
$$

Using the definition of $x^*_{j,k}$ from (9),

$$
d = \sum_{k \in B_j} x^*_{j,k}(1 + r^*_k - e).
$$

Furthermore, by Lemma C.5, for any MFI $i$, there exists a village $j \in V_i$ such that $z^*_{j,i} > 0$. Thus, we get $\mu^*_{j,i} = 0$. The stationary condition gives us

$$
\gamma^*_i = \frac{1}{\sum_{k \in B_j} z^*_{j,k}} > 0.
$$

That is, for each MFI $i$, $\gamma^*_i > 0$. Therefore, $r^*_i > 0$ by (10). Also, (6) holds for $\gamma^*_i > 0$. Again, using the definition of $x^*_{j,i}$ from (9), the other equilibrium condition for our model can be obtained from (6):

$$
T_i - \sum_{j \in V_i} x^*_{j,i} = 0.
$$

Thus, we have the following theorem and corollary.

**Theorem C.7.** *The special case of microfinance markets with identical villages and no loan portfolio diversification, has an equivalent Eisenberg-Gale formulation.*

**Corollary C.8.** *For the above special case, there exists an equilibrium point with unique interest rates [21] and a combinatorial polynomial-time algorithm to compute it [60].*

## C.3 Equilibrium Properties of General Case

**Assumption C.1.**

$$
0 \leq \lambda \leq \frac{1}{2 + \log T_{max}}
$$

*where $T_{max} \equiv \max_i T_i$ and w.l.o.g., $T_i > 1$ for all $i$.*

**Property C.9.** *The first constraint of the village-side optimization program* $(P_V)$ *must be tight at any equilibrium point.*

*Proof.* Suppose that this is not the case, i.e., at an optimal solution $\mathbf{x}_j^*$ for some village $j$, $\sum_{i \in B_j} x_{j,i}^* (1 + r_i - e_j) < d_j$. We will show that village $j$ can improve its objective function by slightly increasing $x_{j,i}^*$ for any $i \in B_j$ while maintaining the constraint. The derivative of the village-side objective function w.r.t. $x_{j,i}$ is

$$1 - \lambda \log x_{j,i}^* - \lambda$$

which is positive (by Assumption C.1 and equilibrium condition $x_{j,i}^* \leq T_i$). $\qquad\square$

We define $e_{max}^i \equiv \max_{j \in V_i} e_j$ and $d_{max}^i \equiv \max_{j \in V_i} d_j$ and obtain the following bounds on interest rates.

**Property C.10.** *At any equilibrium point, for each MFI $i$, $e_{max}^i - 1 < r_i^* \leq \frac{|V_i| d_{max}^i}{T_i} + e_{max}^i - 1$.*

*Proof.* Proof of $e_{max}^i - 1 < r_i^*$ is similar to the proof of Property C.2. Although compared to Property C.2, we have a different objective function here, the proof of Property C.9 shows that increasing $x_{j,i}$ also increases the village objective function.

For the proof of the upper bound, the total amount of loan that villages in $V_i$ can seek from MFI $i$ is at most $\sum_{j \in V_i} \frac{d_j}{1 + r_i - e_j}$ (this bound is obtained using the first constraint in the village-side optimization program $(P_V)$, when each village in $V_i$ seeks loan *only* from MFI $i$). We have the following at an equilibrium point.

$$
\begin{aligned}
T_i &\leq \sum_{j \in V_i} \frac{d_j}{1 + r_i^* - e_j} \\
&\leq d_{max}^i \sum_{j \in V_i} \frac{1}{1 + r_i^* - e_j} \\
&\leq d_{max}^i \frac{|V_i|}{1 + r_i^* - e_{max}^i}
\end{aligned}
$$

Rewriting this, we obtain $r_i^* \leq \frac{|V_i| d_{max}^i}{T_i} + e_{max}^i - 1$. $\qquad\square$

# D  Computational Scheme

## D.1  Computing an Equilibrium Point

---

**Algorithm 1** Outline of Equilibrium Computation

---

1: For each MFI $i$, initialize $r_i$ to $e^i_{max} - 1$.
2: For each village $j$, compute its best response $\mathbf{x}_j$.
3: **repeat**
4:     **for all** MFI $i$ **do**
5:         **while** $T_i \neq \sum_{j \in V_i} x_{j,i}$ **do**
6:             Change $r_i$ as described later.
7:             For each village $j \in V_i$, update its best response $\mathbf{x}_j$ reflecting the change in $r_i$.
8:         **end while**
9:     **end for**
10: **until** no change in $r_i$ occurs for any $i$

---

**Lemma D.1.** *(**Village's Best Response**) Given the interest rates of all the MFIs, the following is the unique best response of any village $j$ to any MFI $i \in B_j$:*

$$x^*_{j,i} = \exp\left(\frac{1 - \lambda - \alpha^*_j(1 + r_i - e_j)}{\lambda}\right) \tag{12}$$

*where $\alpha^*_j \geq 0$ is the unique solution to*

$$\sum_{i \in B_j} \exp\left(\frac{1 - \lambda - \alpha^*_j(1 + r_i - e_j)}{\lambda}\right)(1 + r_i - e_j) = d_j. \tag{13}$$

*Proof.* Following is the Lagrangian of the village-side optimization program $(P_V)$ for village $j$:

$$L(\mathbf{x}_j, \alpha_j) = -\sum_{i \in B_j} x_{j,i} - \lambda \sum_{i \in B_j} x_{j,i} \log \frac{1}{x_{j,i}}$$

$$+ \alpha_j \left(\sum_{i \in B_j} x_{j,i}(1 + r_i - e_j) - d_j\right).$$

At an optimal solution, we have $\frac{\delta L}{\delta x_{j,i}} = 0$ for any $i \in B_j$. This is expanded below:

$$-1 - \lambda \log \frac{1}{x_{j,i}^*} + \lambda x_{j,i}^* \frac{1}{x_{j,i}^*} + \alpha_j (1 + r_i - e_j) = 0$$

$$\Leftrightarrow x_{j,i}^* = \exp \left( \frac{1 - \lambda - \alpha_j (1 + r_i - e_j)}{\lambda} \right).$$

By Property C.9, $\sum_{i \in B_j} x_{j,i}^* (1 + r_i - e_j) = d_j$. Substituting the expression for $x_{j,i}^*$ we obtain the second equation claimed in the statement. Moreover, $\alpha_j^*$ must be unique; otherwise, by the above expression for $x_{j,i}^*$, we would have multiplicity in the best response of village $j$, which is precluded by the convex optimization $(P_V)$. $\qquad\square$

In the above characterization of the village best response, as soon as the interest rate $r_i$ of some MFI $i$ changes in Line 6 of Algorithm 1, both the best response allocation $x_{j,i}^*$ and the Lagrange multiplier $\alpha_j^*$ change in Line 7, for any village $j \in V_i$. Next, we show the direction of these changes.

**Lemma D.2.** *Whenever $r_i$ increases (decreases) in Line 6, $x_{j,i}$ must decrease (increase) for every village $j \in V_i$ in Line 7 of Algorithm 1.*

*Proof.* We prove the case of $r_i$ increasing. The other case can be proved in the same way. First, observe that we cannot simply invoke Equation (12) to prove the statement, because $\alpha_j^*$ has also changed once $r_i$ has changed and the direction of change of $\alpha_j^*$ is not immediately clear from Equation (13).

Here, we treat the terms $r_i$, $x_{j,i}^*$, and $\alpha_j^*$ as *names* of variables instantiated with specific values at each iteration of Lines 6 and 7 of Algorithm 1. Suppose that the value of $r_i$ has been increased in Line 6. Suppose, for a contradiction, that in response to this increase, some village $j \in V_i$ has either increased its value of $x_{j,i}^*$ or kept it unchanged in Line 7. By Property C.9, the first constraint of the village-side program $(P_V)$ is tight at any optimal solution, including village $j$'s previous best response in Line 7 (i.e., the old values of $\mathbf{x}_j$ just before the current update in Line 7). Therefore, in the current best response, village $j$ must decrease the value of $x_{j,k}^*$ for some $k \in B_j$ (otherwise that constraint cannot be satisfied, because $r_i$ has increased). Rewriting the expression of $x_{j,i}^*$ given in Lemma D.1 in terms of $\alpha_j^*$, we obtain the first equation below. The second equation follows similarly.

$$\alpha_j^* = \frac{1 - \lambda - \lambda \log x_{j,i}^*}{1 + r_i - e_j} \tag{14}$$

$$\alpha_j^* = \frac{1 - \lambda - \lambda \log x_{j,k}^*}{1 + r_k - e_j} \tag{15}$$

By Equation (14), our assumption that $x_{j,i}^*$ has increased in response to the increase of $r_i$ implies that the value of $\alpha_j^*$ has decreased from its previous one. Therefore, by Equation (15), the value of $x_{j,k}^*$ must increase, which gives us a contradiction (note that $r_k$ has not been changed, i.e., its value remains the same as the one during village $j$'s previous best response). Therefore, whenever $r_i$ increases in Line 6, $x_{j,i}^*$ must decrease, for all $j \in V_i$. □

The next lemma is a cornerstone of our theoretical results. Here, we use the term *turn of an MFI* to refer to the iterative execution of Line 6, wherein an MFI tries to set its interest rate to make supply equal demand. At the end of its turn, an MFI has successfully set its interest rate to achieve this objective.

**Lemma D.3.** *(**Strategic Complementarity**) Suppose that an MFI $i$ has increased its interest rate at the end of its turn. Thereafter, it cannot be the best response of any other MFI $k$ to lower its interest rate when its turn comes in the algorithm.*

*Proof.* Consider a village $j \in V_i$. By Lemma D.2, when an MFI $i$ increases its interest rate in Line 6, village $j$ must decrease $x_{j,i}^*$ in Line 7. Considering Equation (14), it may at first seem possible that the value of $\alpha_j^*$ can increase, decrease, or even remain the same, depending on how much $x_{j,i}^*$ has decreased. However, we will next show that $\alpha_j^*$ cannot increase. For this, we define $\beta_j^* \equiv \frac{\alpha_j^*}{\lambda}$ and $\rho_{j,i} \equiv 1 + r_i - e_j$ and rewrite Equation (13) as follows.

$$\sum_{i \in B_j} \exp\left(\frac{1 - \lambda}{\lambda}\right) \exp\left(-\beta_j^* \rho_{j,i}\right) \rho_{j,i} = d_j$$

$$\Leftrightarrow \sum_{i \in B_j} \frac{\rho_{j,i}}{\exp\left(\beta_j^* \rho_{j,i}\right)} = d_j \exp\left(\frac{-1 + \lambda}{\lambda}\right)$$

Here, the right hand side is constant, since $\lambda$ and $d_j$ are both constants. Consider the left hand side. It suffices to show that if we increase $\rho_{j,i}$ (i.e.,

increase $r_i$) by *any amount*, but keep $\beta_j^*$ unchanged, then the left hand side must decrease. [7] In this case, only one term of the sum on the left hand side changes: $\frac{\rho_{j,i}}{\exp(\beta_j^* \rho_{j,i})}$. We show that the derivative of this term w.r.t. $\rho_{j,i}$ is non-positive.

$$\frac{1}{\exp(\beta_j^* \rho_{j,i})} - \frac{\beta_j^* \rho_{j,i}}{\exp(\beta_j^* \rho_{j,i})} \leq 0$$
$$\Leftrightarrow \rho_{j,i} \beta_j^* \geq 1$$
$$\Leftrightarrow (1 + r_i - e_j) \frac{\alpha_j^*}{\lambda} \geq 1$$
$$\Leftrightarrow 1 - \lambda - \lambda \log x_{j,i}^* \geq \lambda, \quad \text{by Equation (14)}$$
$$\Leftrightarrow \lambda \leq \frac{1}{2 + \log x_{j,i}^*}$$

which holds by Assumption C.1. Therefore, $\alpha_j$ cannot increase when $r_i$ increases.

Since $\alpha_j$ can only decrease when $r_i$ increases, using Equation (15) we obtain that in Line 7 of the algorithm, village $j$ cannot decrease $x_{j,k}^*$ for any $k \neq i \in B_j$. Thus, when its next turn comes, MFI $k$ can only find a rise in demand for its loans, which can only exceed $T_k$, since at the end of every turn, an MFI successfully sets its interest rate so that the demand for its loan equals its supply. Therefore, by Lemma D.2, decreasing its interest rate cannot be MFI $k$'s best response. $\qquad\square$

In essence, Lemma D.2 is a result of *strategic substitutability* [20] between the MFI and the village sides, while Lemma D.3 is a result of *strategic complementarity* [16] among the MFIs. We will see that our algorithm exploits these two properties as we fill in the details of Lines 6 and 7 next.

### D.1.1   Line 6: MFI's Best Response

By Lemma D.2, the total demand for MFI $i$'s loan monotonically decreases with the increase of $r_i$. Therefore, a simple search, such as a binary search, between the upper and the lower bounds of $r_i$ as stated in Property C.10, can efficiently find the "right" value of $r_i$ that makes supply equal demand for $i$. For example, in the first iteration of the while loop, in Line 6, $r_i$ is set to the midpoint $r_i^m = \frac{r_i^l + r_i^h}{2}$ between its lower bound $r_i^l$ and upper bounds $r_i^h$ . Then the best response of the villages are computed in the next

line. If still $T_i \neq \sum_{j \in V_i} x_{j,i}$ then in the next iteration, in Line 6, $r_i$ is set to either $\frac{r_i^l + r_i^m}{2}$ or $\frac{r_i^m + r_i^h}{2}$ depending on whether $T_i > \sum_{j \in V_i} x_{j,i}$ or the opposite, respectively. The search progresses in this way until $T_i = \sum_{j \in V_i} x_{j,i}$. As an implementation note, to circumvent issues of numerical precision, we can adopt the notion of $\epsilon$-*equilibrium point*, where the market $\epsilon$-clears (i.e., the absolute value of the difference between supply and demand for each MFI $i$ is below $\epsilon$) and each village plays its $\epsilon$-best response (i.e., it cannot improve its objective function more than $\epsilon$ by changing its current response). Having said that, all of our results hold for $\epsilon = 0$.

### D.1.2   Line 7: Village's Best Response

We use Lemma D.1 to compute each village $j$'s best response $x_{j,i}^*$ to MFIs $i \in B_j$. However, Equation (12) requires computation of $\alpha_j^*$, the solution to Equation (13). We can exploit the convexity of the left hand side of Equation (13) to design a simple search algorithm to find $\alpha_j^*$ up to a desired numerical accuracy.

Next, we make the following statement about our constructive proof of the existence of an equilibrium point.

**Theorem D.4.** *There always exists an equilibrium point in a microfinance market specified by programs ($P_M$) and ($P_V$).*

*Proof.* Algorithm 1 begins with initial values of interest rates arbitrarily close to their lower bound established in Property C.10. Thereafter, by Lemma D.3, these interest rates can only increase, and by Lemma D.1, every village has a unique best response to these interest rates. Now, the interest rates are upper bounded by Property C.10. Therefore, by the well-known monotone convergence theorem, the process of incrementing the interest rates must come to an end. And that point of termination must be an equilibrium point. □

## D.2 More on Parameter Learning

$$\min_{\mathbf{e},\mathbf{d},\mathbf{r}} \sum_i \sum_{j\in V_i} (x_{j,i}^* - \tilde{x}_{j,i})^2 + C \sum_i (r_i^* - \tilde{r}_i)^2$$

such that

for all $j$,

$$\mathbf{x}_j^* \in \arg\max_{\mathbf{x}_j} \sum_{i\in B_j} x_{j,i} + \lambda \sum_{i\in B_j} x_{j,i} \log \frac{1}{x_{j,i}}$$

$$\text{s. t. } \sum_{i\in B_j} x_{j,i}(1 + r_i^* - e_j) \leq d_j$$

$$\mathbf{x}_j \geq 0 \tag{16}$$

$$\mathbf{e}_j \geq 1, \ \mathbf{d}_j \geq 0$$

$$\sum_{j\in V_i} x_{j,i}^* = T_i, \text{ for all } i$$

$$r_i \geq e_j - 1, \text{ for all } i \text{ and all } j \in V_i$$

Initialization plays a big role in solving this problem fast, especially in large instances (e.g., the instance with data from Bangladesh that has thousands of constraints and variables). If we initialize the parameters arbitrarily, then the interior point algorithm spends an enormous amount of time searching for a feasible solution. Fortunately, we can avoid this issue by computing a feasible solution first. For this, with arbitrary values of $\mathbf{e}$ and $\mathbf{d}$ as inputs, we run Algorithm 1 and compute an initial equilibrium point with respect to the inputs of $\mathbf{e}$ and $\mathbf{d}$. Subsequently, the learning procedure updates $\mathbf{e}$ and $\mathbf{d}$ compute an optimal solution. Using such an initial feasible solution to the above optimization problem, we observed a much faster convergence.

In the next section, we will show that the above learning procedure does not overfit the real-world data. We will also highlight the issue of equilibrium selection for parameter estimation.

# E   Empirical Study

Our empirical study is based on microfinance data of Bangladesh and Bolivia. The reason we have chosen these two countries is that over time, microfinance programs in these two countries have behaved very much differently with respect to competition and interest rates [51].

### E.1 Case Study: Bolivia

#### E.1.1 Data

We have obtained microfinance data of Bolivia from several sources, such as ASOFIN, [8] the apex body of MFIs in Bolivia, and the Central Bank of Bolivia. [9] We were only able to collect somewhat coarse, region-level data. The data, dated June 2011, consists of eight MFIs operating in 10 regions. These MFIs (and their interest rates) are: Bancosol (21.54%), Banco Los Andes (19.39%), Banco FIE (20.49%), Prodem (23.55%), Eco Futuro (29.25%), Fortaleza (21.22%), Fassil (22.38%), and Agro Capital (21%).The number of edges in the bipartite network is 65, out of a maximum possible 80.

#### E.1.2 Learning the Parameters of the Model

Given the exogenous parameter $\lambda$, the learning scheme above estimates the parameters $e_j$ and $d_j$ such that an equilibrium point of the game is a close approximation of the observed data. Let us first explain how we choose the exogenous diversification parameter $\lambda$.

Figure 1 shows how the objective function of the optimization program varies as a function of $\lambda$. It shows that for a range of smaller values of $\lambda$, the objective function value of the learning program is consistently small (note that the optimization routine wants to minimize the objective function). As $\lambda$ grows, the objective function value oscillates a lot and is sometimes very high.

Also shown in Figure 1 is how the interest rates become dissimilar as $\lambda$ is varied. For that, we first define a *negative entropy* term, $C\sum_i \frac{r_i}{Z}\log\frac{Z}{r_i}$, where $Z = \sum_i r_i$, where $C$ is a constant set to 100. Here, the negative entropy quantifies the similarity among the interest rates. That is, high negative entropy means the interest rates among the MFIs are more similar. As we can see, at relatively low values of $\lambda$, the interest rates are similar to each other and as $\lambda$ becomes high, they become very much dissimilar at some points.

We choose $\lambda = 0.05$, because at this level of $\lambda$, the objective function value of the learning optimization is low as well as stable and the interest rates are also allowed to be relatively dissimilar. As we will show later, dissimilarities among the interest rates of the MFIs are very often observed in the real-world data.

Figure 1: *Optimal objective function values and "negative entropy" of interest rates as λ varies. This shows that the objective function value becomes large (which is undesirable) as λ grows.*

The learned values of the parameters $e_j$ and $d_j$ for villages $j$ in the Bolivia market capture the variation among the villages with respect to the revenue generation function. Although the rate $e_j$ of revenue generation varies only from 1.001 to 1.234 among the villages, the variation in $d_j$ is much greater.

The individual loan allocations learned from data closely approximate the observed allocations. In fact, the average relative deviation between these two allocations is only 4.41% (relative deviation is calculated by $\frac{\text{abs}(x_{j,i} - \tilde{x}_{j,i})}{\sum_i \tilde{x}_{j,i}}$). Figure 2 shows this. The $45^o$ line is the locus of equality between these two allocations.

Figure 2: *Learned allocations vs. observed allocations. This shows that the learned allocations closely approximate the allocations in the observed data.*

The learned model matches the total loan allocations of the MFIs due to the constraint of the program. As shown in Figure 9, the learned interest

rates are, however, slightly different from the observed rates.

**Issues of Bias and Variance.**

Our dataset consists of a single sample. As a result, the traditional approach of performing cross validation using hold-out sets or plotting learning curves by varying the number of samples would not work in our setting. To investigate whether our model overfits the data, we have applied the following procedure of systematically introducing noise to the observed data sample. In the case of overfitting, increasing the level of noise would lead the equilibrium outcome to be significantly different from the observed data.

We use a parameter $\nu$ to control the level of noise. For any fixed $\nu$, we derive noisy samples by modifying each non-zero observed allocation $\tilde{x}_{j,i}$ by adding to it a random noise (under certain noise models to be described later). We denote the resulting noisy allocation by $x^{\nu}_{j,i}$ and treat the newly constructed noisy dataset as a training set and the observed dataset as test. We learn the parameters of the model using the training set. We then find an equilibrium allocation $\mathbf{x}^*$ using Algorithm 1. [10] We compute the following mean relative deviation as the test error: $\frac{1}{n}\sum_i \frac{1}{|V_i|}\sum_j \frac{\text{abs}(x^*_{j,i}-\tilde{x}_{j,i})}{T_i}$. The training error is computed similarly (by replacing $\tilde{x}_{j,i}$ by $x^{\nu}_{j,i}$). For each noise level $\nu$, we perform the whole procedure a number of times and calculate the average error.

**Gaussian Noise Model.** In this model, we obtain $x^{\nu}_{j,i}$ by adding to each non-zero observed allocation $\tilde{x}_{j,i}$ a Gaussian random noise of mean 0 and standard deviation $\nu\sigma(i)$, where $\sigma(i)$ is the standard deviation of the allocations of MFI $i$ across all villages in which it operates. For the Bolivia dataset, we find that varying the noise level $\nu$ between 0 and 1 and taking the average over 25 trials, both the training and test errors are below 5.83% and are close to each other (i.e., within the 95% confidence interval of each other). The learning curve, shown in Figure 3, does not suggest overfitting.

**Dirichlet Noise Model.** In this noise model, we derive noisy allocations while keeping the total amount of loan disbursed by each MFI the same as its observed total amount. We follow the commonly used procedure of deriving a Dirichlet distribution from a gamma distribution [27, Ch. 18]. We control the noise (i.e., variance) of the Dirichlet distribution using the parameter $\nu$ in the following way. For each MFI $i$, $\mathbf{x}^{\nu}_i = T(i) \times \text{Dir}(\nu\tilde{\mathbf{x}}_i)$. [11] As the $\nu > 0$ increases, the variance of the distribution $\text{Dir}(\nu\tilde{\mathbf{x}}_i)$ decreases.

Figure 3: *Learning curve for the Bolivia dataset under Gaussian random noise (vertical bars denote 95% CI). The learning curve does not suggest overfitting.*

Varying $\nu$ from $2^{-5}$ (high variance) to $2^{15}$ (low variance) and taking the average over 50 trials at each $\nu$, we found that the training and the test errors are within the confidence intervals of each other across the whole spectrum of noise levels. The maximum test error of 8.83% occurs at $\nu = 2^{-5}$ where we also get the maximum offset of the 95% confidence interval, which is 0.66%. Once again, the learning curve, shown in Figure 4, does not suggest overfitting.

Figure 4: *Learning curve for the Bolivia dataset under the Dirichlet noise model. Logarithm of the noise parameter $\nu$ is shown on the x-axis (vertical bars denote 95% CI). The learning curve in this noise model does not indicate overfitting as well.*

**Equilibrium Selection.**

In practice, equilibrium selection is an important issue. In general, we cannot rule out the possibility of multiplicity of equilibria. In such cases, our learning scheme biases its search for an equilibrium point that most closely explains the data. One important question is: does the equilibrium point that we compute change drastically when noise is added to our data? In other words, how robust is our scheme? To answer this, we extend the above experimental procedure using the following bootstrapping scheme. [12]

Suppose that for each noise level $\nu$, we have $t$ trials (i.e., $t$ noisy training sets, each derived from the observed dataset using a particular noise model with the given parameter $\nu$). For each noise level $\nu$, we iterate the following procedure $M$ times. At each iteration $k$, we uniformly sample $t$ times (with replacement, of course) from the $t$ noisy training sets and then compute the following relative mean equilibrium allocations: $\hat{\mu}_{j,i} = \frac{1}{t} \sum_{l=1}^{t} \frac{x_{j,i}^{*(l)}}{T_i}$ (here, $x_{j,i}^{*(l)}$ denotes equilibrium allocation for the $l$-th training set). Within the same $k$-th iteration, we compute the following average deviation from mean: $\hat{\delta}(k) = \frac{1}{n} \sum_i \frac{1}{|V_i|} \sum_j \frac{1}{t} \sum_l \mathrm{abs}(x_{j,i}^{*(l)}/T_i - \hat{\mu}_{j,i})$. This quantity signifies the average distance of the equilibria of the sampled examples from the mean equilibrium. Now, for each value of $\nu$, we average this distance measure over these $M$ iterations. We perform this bootstrapping procedure for various values of $\nu$.

Figure 5: *Average deviation of the equilibrium points from the mean for the Bolivia dataset under the Gaussian noise model (vertical bars denote 95% CI). It shows that the equilibrium point computed is robust with respect to noise.*

Figure 6: *Average deviation of the equilibrium points from the mean for the Bolivia dataset under the Dirichlet noise model. Logarithm of the noise parameter $\nu$ is shown on the x-axis. Vertical bars denote 95% CI. It also shows the robustness of the computed equilibrium point although the noise model is different—Dirichlet.*

For the Bolivia dataset, under the Gaussian noise model (described above) and using $t = 25$ and $M = 100$, we found that this average distance varies from 0.79% to 0.96%, with the offset of the 95% CI ranging from 0.015% to 0.026% for varied noise levels $0 < \nu \leq 1$. On the other hand, for the Dirichlet noise model and using $t = 50$ and $M = 100$, the maximum average distance is 6.35%, which happens at a very high variance parameterized by $\nu = 2^{-5}$. The minimum average distance of 0.10% happens at low variance with $\nu = 2^{14}$. The offset of the 95% CI ranges from 0.001% to 0.05% across all the noise levels considered. The plots for these two noise models are shown in Figures 5 and 6. Moreover, under both noise models, the equilibrium interest rates do not deviate much from the mean either. These suggest that an equilibrium point does not change much when noise is introduced to the data and that our scheme is robust with respect to noise in the real world data.

### E.1.3 Equilibrium Computation

We discuss equilibrium computation on the model learned with $\lambda = 0.05$. First, we would like to remind the reader about the point we made regarding equilibrium selection. In practice, we have observed that the equilibrium computed by Algorithm 1 converges to the learned values of $\mathbf{x}$ and $\mathbf{r}$, *even if we start with different initial values*. For example, Figure 7 shows the case of MFI Bancosol's convergence to the same equilibrium interest rate despite different initialization (other interest rates were also differently initialized).

This equilibrium interest rate is the same as the learned one. Not only that, as Figure 8 shows individual loan allocations were also almost the same.

Figure 7:    *Two best response dynamics of MFI Bancosol with different initialization. Both of these converged to the same solution.*

Figure 8:    *Learned allocations vs. equilibrium allocations. The two allocations being almost identical, shows that the learning algorithm was able to capture an equilibrium point using the inner part of the nested optimization program.*

Finally, Figure 9 shows a comparison among the observed, learned, and equilibrium interest rates.

Figure 9: *Comparison among observed, learned, and equilibrium interest rates. The equilibrium interest rates and the observed interest rates do not completely match, which is fine as we do not assume the data to be an equilibrium point.*

## E.2 Case Study: Bangladesh

### E.2.1 Data

We have obtained microfinance data, dated December 2005, from Palli Karma Sahayak Foundation (PKSF), which is the apex body of NGO MFIs in Bangladesh. There are seven major MFIs (or collection of MFIs) operating in 464 *upazillas* or collection of villages. The data can be simplified as a 464-by-7 matrix where an element in location $(j, i)$ denotes the number of borrowers that MFI $i$ has in village $j$. The bipartite network-structure induced by this data is very dense, consisting of 3096 edges out of a maximum possible 3248.

The seven major MFIs or bodies of MFIs (and their flat interest rates) are BRAC (15%), ASA (15%), PKSF partner organizations (12.5%), Grameen Bank(10%), BRDB (8%), Other government organizations (8%), and Other MFIs (12.5%) [51, 61].

### E.2.2 Learning the Parameters of the Model

Due to the size of Bangladesh data, we are posed with the problem of solving a nonlinear optimization problem of the order of thousands of variables and constraints. As discussed above, the interior point algorithm is initialized with a feasible solution, which makes computation much faster. Still, solving the problem takes time in the order of hours, compared to minutes for the Bolivia case.

Similar to the Bolivia case, the learned parameters $e_j$ (rate of revenue generation) and $d_j$ (revenue from other sources) show variation among the villages $j$. This is particularly the case with the estimated parameter $d_j$, while the estimation of $e_j$ varies around 1.07 for all the villages. A more detailed analysis of the estimated parameters (for example, their correlation with access to resources such as rivers) is left for future work.

We also obtain a close approximation of observed individual allocations in the learned model (see Figure 10. For example, the average deviation is only 5.54% when $\lambda = 0.05$. The market clears in the learned model, and as shown in Figure 11, the learned interest rates are close to the actual ones, except for the government MFIs numbered 5 and 6, which are known to be operating inefficiently, i.e., with much lower interest rates (8%) than that required for sustainability without subsidies [18, 61].

Figure 10: *Learned allocations vs. observed allocations. Although they are not exactly the same, the learned allocations do approximate the observed ones.*

### E.2.3  Equilibrium Computation

Similar to the Bolivia case, we have observed that the best response dynamics of Algorithm 1 quickly converges to the allocations and interest rates of the learned model. Figures 12 shows the similarity between learned and equilibrium allocations.

## Footnotes

[1] Equation (2) shows that the best response of a player depends on the *difference* between payoff functions and hence on the difference between the corresponding random parts of Equation (1). Along with other simplifying assumptions, Bjorn and Vuong's main assumption is that this difference between the random parts is a standard normal distribution with a correlation between the two players.

[2]For example, the husband's reaction function could be one of the followings: choosing action 1 all the time (no matter what the wife has chosen), choosing 0 all the time, choosing whatever the wife has chosen, and choosing the opposite of what the wife has chosen.

[3]It is important to note that in the above argument, the village side has been allowed

[5]This is not *exactly* the convex program that Eisenberg and Gale defined [21, p. 166], but rather a simple variant of that.

[6]The quantities at an optimal solution are denoted by $^*$.

[7]Note that increasing $\beta_j^*$ will only further decrease the left hand side.

[8]http://www.asofinbolivia.com

[9]http://www.bcb.gob.bo/

[10]Note that this equilibrium allocation would have remained the same all the time had we treated the the observed dataset as the training set.

[11]Slightly abusing the notation, the vector $\tilde{\mathbf{x}}_i$ corresponds to non-zero observed allocations only.

[12]In the previous experiment on the issue of overfitting, the focus was on the distance between an equilibrium point and the data. Now, our focus is on the distance between different equilibrium points when noise is added to the data.