[Reviews · NeurIPS 2014]

Submitted by Assigned_Reviewer_6

The authors model the interactions of a microfinance market as a game between the banks and the villages [typo lenders]. They show that such interactions have a market-clearing equilibrium. They apply this model by looking at microfinance data and fitting the data to a nearby equilibrium solution. The learned parameters are then utilized in predicting how the market would perform under perturbations.

Equilibrium analysis for microfinance appears to be the main contribution of the paper (such analysis is known in other situations like paid search auctions) [edit for clarity: inference from the computed equilibrium is what I mean by analysis here]. The main problem, however, is that the paper is not in a publishable state and needs a re-write to make it readable. For instance, the entire introduction section only talks about microfinance in general. The sections abruptly ends with a justification for a few assumptions that leave the reader bewildered as to the connection with the work.
Summary: The results seem interesting but the paper needs a lot of work in flow and readability.

Submitted by Assigned_Reviewer_9

The authors present a novel application of statistical learning and optimization, by algorithmically computing an equilibrium point, to the problem of interventions in networked microfinance economies.

The paper is very well written and presents the problem, as well the solution, very clearly. The authors present a simple algorithm for computing the market-clearing interest rates in equilibrium. The only issue is that from the paper, it is not clear whether the equilibrium has any social welfare properties, for e.g. are the interest rates reasonable?

I think this paper deserves to be presented at NIPS. The community would benefit from being introduced to this type of data, that the paper presents clearly, that can result in more people looking at the problem with other approaches.
Summary: This is an application paper that is very well-written and easy to understand, that introduces an important problem of interventions in networked economies. The NIPS community would benefit from being introduced to this type of data.

Submitted by Assigned_Reviewer_33

SUMMARY: This paper presents a model for analyzing policy decisions, particularly ones that require some causal knowledge, in microfinance markets. The model is justified by several studies, and analyzed to show the existence of equilibrium prices. A computational framework is then presented to compute an equilibrium and learn the parameters of the model from real micofinance data. Finally, the authors leverage the framework to provide answers to several policy questions.

CRITIQUE: By all metrics, this is a good paper. It is clear, well-motivated, well-supported by analysis and data, introduces novel techniques, and has a potential for impact. I point out a few specific problems below, and I do wonder whether this is a good fit for NIPS, but defer to more senior members for the latter.

SUGGESTION: The assumption that MFIs are non-profits is crucial to the entire analysis, yet in my mind is a source for some of the more interesting questions. At the very least, it would be interesting to note what changes when MFIs are profit seeking, either analytically or empirically. Specifically, could the social welfare actually increase if MFIs were even slightly profit seeking, and therefore requiring fewer subsidies over time? It seems that if you expanded your model slightly you might be able to answer this question.

SPECIFIC COMMENTS:

page 2:
. "[13] and joint-liability contracts would mitigate the risks of adverse selection [13]" -- odd to cite [13] twice in the same sentence
. "study of causality" -- this term is not really defined, and to many people may seem extremely vague (i.e. all of science); even though you very nicely explain everything in the appendix, it might go a long way to describe the scope of "causality" here
. "Put differently, what would be a game-theoretic analog of the do operation [21, p. 23] used for surgeries in probabilistic settings?" -- This sentence is somewhat meaningless unless you know the reference; please clarify the context

page 3:
. "subset of the village"[s]
. "interest rate at which MFI i gives loan" -- the phrase "gives loan" seemed odd to me
. "max_{r_i} 1" -- this is jarring without an explicit reassurance that you indeed intended max 1
. "each MFI i is optimizing (P_M)" -- add "the left-hand side of" (one cannot optimize an inequality)

page 4:
. "we model the village side as non-corporate agents." -- Where does this come up in the model?
. Property 3.2. -- Interpret this for us in words
. "trivial allocations" -- Again, remind us what x is by saying why this would be trivial

page 5:
. "that a more restricted case" -- this sentence has a bug
. "Change ri as described later." -- Cite a specific equation number so the reader isn't left looking for where you fill this in (it's not obvious)

page 6:
. "Our model computes lower equilibrium interest rate"
Summary: Clear, well-motivated, well-supported by analysis and data, introduces novel techniques, and has a potential for impact.
Author Feedback
Author rebuttal: Assigned_Reviewer_33
------------------------------
We thank the reviewer for a very detailed review with many specific points. The reviewer’s CRITIQUE is very well taken. The reviewer’s SUGGESTION to study the consequences of allowing profit-making MFIs in our model is scientifically interesting and certainly presents a very promising future direction.

Regarding the reviewer’s comment on relevance, the reviewer might be unaware that one of the main subject areas in this year’s NIPS is computational economics (http://nips.cc/Conferences/2014/PaperInformation/Keywords). We believe that our paper fits well in that category.

Interpretation of Property 3.2: At any equilibrium point, there is a lower bound on the interest rate of every MFI. This lower bound can be expressed in terms of the revenue generation rates of the villages that are served by the corresponding MFI. The algorithm for equilibrium computation uses this lower bound.

Implication of “we model the village side as non-corporate agents:” The village side does not have the goal of maximizing its net revenue.

Following are the possible textual changes as per the reviewer’s suggestions (we also fixed the typos). We would appreciate any feedback. We verified that the page limit would not be exceeded.

1. To clarify the “the study of causality” (page 2), we could qualify it as follows:
“… the study of causality, which studies cause and effect questions using a mathematical model of a real-world phenomenon, …”
2. “do operation” (page 2): We could replace the two sentences “Put differently … by changing the ‘structure’ of the game” by the following sentence.
“Analogous to the probabilistic settings [21], the types of surgeries we consider in this paper change the ‘structure’ of the game.”
3. Page 3: “interest rate at which MFI i gives loan” --> “interest rate of MFI i”
4. Page 3: “max_{r_i} 1”--> Just before the program (P_M), we could add, “Here, the objective function is a constant due to the MFIs’ goal of market-clearance.”
5. Page 4: “trivial allocations” – we could qualify it as follows: “trivial allocations such as all the allocations being zero …”
6. “Change r_i as described later.” --> “Change r_i as described after Lemma 4.3.”

Assigned_Reviewer_6
------------------------------
We thank the reviewer for the review. Below are some clarifications.

1. We modeled the interactions between banks and villages (or borrowers), not between banks and lenders.

2. Equilibrium analysis was not our main contribution. Modeling a microfinance market, learning the model from data, computing equilibrium, and performing causal inference by interventions were the main contribution. As a result, the reviewer’s comment on similarities with paid search auctions is a little unclear to us.

3. Regarding the reviewer’s main concern about readability and flow, we tried to ensure these within the space allowed. For example, in page 4, before the statement of each property we mentioned what we were going to do and we followed that statement by a proof sketch. However, as the reviewer pointed out, we kept the introduction brief (for example, we did not end the introduction by saying what we were going to do in the upcoming sections). Our intent was to accommodate the technical content of the paper (e.g., all the proof sketches) within the limited space.

Assigned_Reviewer_9
------------------------------
We thank the reviewer for a detailed review. We are uncertain as to how to answer the reviewer’s question about social-welfare properties. However, we do want to make some observations regarding “social welfare.”

First, note that we have two types of players, MFIs and villages. So the concept of social welfare is unclear and at least relative in our model.

Second, from a theoretical standpoint, if we define social welfare as the sum of the utilities of the villages (or even all the players, because the MFIs’ utilities are constant), then the equilibrium condition suggests that, as the level of diversification decreases, every equilibrium allocation becomes social-welfare optimal because it would equal the total amount of loan money available across the whole system. The equilibrium properties are unclear for other possible definitions of social welfare (e.g., maximizing the minimum utility over all the villages or minimizing the maximum interest rate over all MFIs, at equilibrium). Understanding such properties would be an interesting future direction.

Third, from an empirical perspective, we did not assume that the real-world interest rates are indeed equilibrium interest rates. In one dataset, after learning the model from the data and computing an equilibrium point, we found that the equilibrium interest rates are in general lower than the real-world interest rates. Interestingly, the central governing body of the MFIs later introduced an interest rate cap, which lowered the real-world interest rates. Also, that interest rate cap is almost the same as the one computed by our method.

In brief, there are some theoretical results and empirical evidence for some types of social-welfare properties at equilibrium, but the full understanding would require further research. We thank the reviewer for bringing up this point.